# Toward Efficient Kernel-Based Solvers for Nonlinear PDEs

**Zhitong Xu** [* 1]  **Da Long** [* 1]  **Yiming Xu** [2]  **Guang Yang** [1]  **Shandian Zhe** [1]  **Houman Owhadi** [3]

## Abstract

We introduce a novel kernel learning framework toward efficiently solving nonlinear partial differential equations (PDEs). In contrast to the state-of-the-art kernel solver that embeds differential operators within kernels, posing challenges with a large number of collocation points, our approach eliminates these operators from the kernel. We model the solution using a standard kernel interpolation form and differentiate the interpolant to compute the derivatives. Our framework obviates the need for complex Gram matrix construction between solutions and their derivatives, allowing for a straightforward implementation and scalable computation. As an instance, we allocate the collocation points on a grid and adopt a product kernel, which yields a Kronecker product structure in the interpolation. This structure enables us to avoid computing the full Gram matrix, reducing costs and scaling efficiently to a large number of collocation points. We provide a proof of the convergence and rate analysis of our method under appropriate regularity assumptions. In numerical experiments, we demonstrate the advantages of our method in solving several benchmark PDEs. Our implementation is released at `https://github.com/BayesianAIGroup/Efficient-Kernel-PDE-Solver`.

## 1 Introduction

Solving partial differential equations (PDEs) stands as a central task in scientific and engineering domains. Recently, machine learning (ML)-based solvers have garnered significant attention. Unlike traditional numerical methods, ML-based solvers eliminate the need for complex mesh designs and intricate numerical techniques, enabling simpler, faster, and more convenient implementation and use. Among these solvers, kernel methods or Gaussian processes (GPs) (Williams & Rasmussen, 2006) hold promise due to their solid mathematical foundations, offering high expressiveness, robustness, and the ability to quantify and reason under uncertainty. Recently, Chen et al. (2021) introduced a general kernel method to approximate solutions of nonlinear PDEs. They augmented the representation by incorporating differential operators (more generally, linear operators) into the kernels, and jointly estimate the solution values and their derivatives of the PDE on a set of collocation points. The approach has shown promising performance in solving several benchmark nonlinear PDEs, backed by a rigorous error analysis, including both convergence and convergence rates (Chen et al., 2021; Batlle et al., 2023).

Despite its success, the methodology requires manual construction of a Gram matrix between the solution and its derivatives that show up in the PDEs. It enumerates combinations between the operators on the kernel to compute different sub-blocks. The process enlarges the size of the Gram matrix (as compared to the number of collocation points), and the computation becomes challenging with a large number of collocation points, a crucial factor for capturing complex PDE solutions such as those with potential roughness (or even non-smoothness) and higher frequencies.

In response, this work proposes an alternative kernel-based framework for solving nonlinear PDEs with several key contributions:

- **Framework**: We remove the basis functions associated with the differential evaluation functionals in the approximation and use a standard kernel interpolation for solution modeling. To approximate solution derivatives, we directly differentiate the interpolant. By minimizing the RKHS norm along with a boundary and residual loss, we estimate the solution without the need for manually constructing a complex Gram matrix. Implementation is straightforward and convenient with the aid of modern automatic differential libraries.

- **Computational Method**: Our framework allows an immediate use of many existing efficient Gram matrix computation and/or approximation techniques. As an instance, we propose to place collocation points on a grid and employ a product kernel over each input

---

*Equal contribution [1]University of Utah [2]University of Kentucky [3]California Institute of Technology. Correspondence to: Shandian Zhe <zhe@cs.utah.edu>.

*Proceedings of the 42nd International Conference on Machine Learning*, Vancouver, Canada. PMLR 267, 2025. Copyright 2025 by the author(s).

dimension. This choice induces a Kronecker product structure within both the kernel interpolation and its differentiation, bypassing the need for computing the entire Gram matrix. Such modification results in a substantial reduction in computational costs, enabling efficient processing of tens of thousands or even millions of collocation points. This is achieved without any sparse and/or low-rank approximations.

- **Theorem:** We provide a rigorous analysis of our framework. We show the convergence and convergence rate of our method under appropriate PDE stability and regularity assumptions that are similar to the assumptions used in the prior work (Batlle et al., 2023). However, our results are not a trivial extension of the prior work in that while our framework uses a reduced model space for efficient computation, our convergence results are as comparably strong as those in the prior work (Chen et al., 2021; Batlle et al., 2023) that employs a richer model space. This is achieved through a more sophisticated proof. We construct an interpolate of the true solution as an intermediate bridge connecting the true solution and our approximation, via which we prove our learning objective is not only feasible, the learned approximation also has a bounded RKHS norm. Next, via using domain decomposition, sampling inequality and mean inequality, we are able to bound the $L_2$ norm of the error w.r.t the PDE operators. Combined with the bounded RKHS norm of our approximation, we establish convergence and convergence rate. The results theoretically affirm the efficacy of our method in yielding accurate solutions.

- **Experiments**: Evaluation on Burgers', nonlinear elliptical, Eikonal, and Allen-Cahn equations have demonstrated our method's practical efficacy. For less challenging scenarios where a small number of collocation points, *e.g.*, 1000, is sufficient, our method achieves comparable or sometimes smaller errors than the existing methods. In more challenging scenarios, such as Burgers' equation with a viscosity of 0.001, our method seamlessly scales to tens of thousands of collocation points, yielding low errors on the order of $10^{-3}$ to $10^{-6}$, underscoring its robustness and accuracy across a range of problem complexities.

## 2 Background

Consider a PDE of the general form,

$$\mathcal{P}(u) = f(\mathbf{x}) \ (\mathbf{x} \in \Omega), \quad \mathcal{B}(u) = g(\mathbf{x}) \ (\mathbf{x} \in \partial\Omega), \quad (1)$$

where $\mathbf{x} = (x_1, \ldots, x_d)^\top$, and $\mathcal{P}$ and $\mathcal{B}$ are nonlinear differential operators in the interior $\Omega$ and boundary $\partial\Omega$, respectively. We assume $\mathcal{P}$ and $\mathcal{B}$ are composed by a series of linear operators, namely,

$$\mathcal{P}(u)(\mathbf{x}) = P\left(L_1(u)(\mathbf{x}), \ldots, L_{Q_\Omega}(u)(\mathbf{x})\right), \ \mathbf{x} \in \Omega,$$
$$\mathcal{B}(u)(\mathbf{x}) = B\left(L_{Q_\Omega+1}(u)(\mathbf{x}), \ldots, L_Q(u)(\mathbf{x})\right), \ \mathbf{x} \in \partial\Omega,$$

where $P(\cdot)$ and $B(\cdot)$ are nonlinear functions, each $L_j$ ($1 \leq j \leq Q$) is a linear operator, such as $\partial_{x_1 x_1} u$, $\partial_{x_1 x_2} u$, $a \cdot \partial_{x_2 x_2} u + b \cdot u$, *etc.*

To solve the PDE (1), Chen et al. (2021) proposed to sample a set of collocation points, $\mathcal{M} = \{\mathbf{x}_1, \ldots, \mathbf{x}_{M_\Omega} \in \Omega, \mathbf{x}_{M_\Omega+1}, \ldots, \mathbf{x}_M \in \partial\Omega\}$, and estimated the solution values and all the relevant linear operators over the solution, $\{L_j(u)(\cdot)\}_j$, evaluated at the collocation points. Specifically, a nested optimization problem was formulated as

$$\underset{\mathbf{z}}{\text{minimize}} \begin{cases} \underset{u \in \mathcal{U}}{\text{minimize}} & \|u\|_\mathcal{U} \\ \text{s.t.} & L_j(u)(\mathbf{x}_m) = z_m^j, \\ & 1 \leq m \leq M_\Omega \text{ for } j = 1 \ldots Q_\Omega, \\ & M_\Omega + 1 \leq m \leq M \text{ for } j = Q_\Omega + 1 \ldots Q \end{cases}$$

$$\text{s.t. } P(z_s^1, \ldots, z_s^{Q_\Omega}) - f(\mathbf{x}_s) = 0 \ (1 \leq s \leq M_\Omega),$$
$$B(z_q^{Q_\Omega+1}, \ldots, z_q^Q) - g(\mathbf{x}_q) = 0 \ (M_\Omega + 1 \leq q \leq M),$$

where $\mathcal{U}$ is the Reproducing Kernel Hilbert Space (RKHS) associated with a kernel $\kappa(\cdot, \cdot)$. Let us denote all $\{z_m^j\}$ as $\mathbf{z}$. With the RKHS $\mathcal{U}$, the similarity (or covariance) between any $z_{m_1}^{j_1}$ and $z_{m_2}^{j_2}$ in $\mathbf{z}$ is

$$c(z_{m_1}^{j_1}, z_{m_2}^{j_2}) = L_{j_1} \circ L_{j_2}(\kappa)(\mathbf{x}_{m_1}, \mathbf{x}_{m_2}),$$

where $L_{j_1}$ is applied along the first argument of $\kappa(\cdot, \cdot)$ and $L_{j_2}$ is applied along the second argument, and then we evaluate at $\mathbf{x}_{m_1}$ and $\mathbf{x}_{m_2}$. For example, suppose $L_1(u) = \partial_{x_1} u$ and $L_2(u) = \partial_{x_2} u$, then $L_1 \circ L_2(\kappa)(\mathbf{x}, \mathbf{x}') = \partial^2 \kappa(\mathbf{x}, \mathbf{x}')/\partial x_1 \partial x_2'$ where $x_1$ and $x_2'$ are the first and second elements of the inputs $\mathbf{x}$ and $\mathbf{x}'$, respectively.

The true solution $u^*$ is assumed to reside in $\mathcal{U}$, and $\|\cdot\|_\mathcal{U}$ is the RKHS norm associated with $\mathcal{U}$. From the representation theorem (Owhadi & Scovel, 2019), it is straightforward to show that the optimum of the aforementioned nested optimizatin problem, denoted by $v_M$, takes the form

$$v_M(\mathbf{x}) = c(v_M(\mathbf{x}), \mathbf{z})\mathbf{C}(\mathbf{z}, \mathbf{z})^{-1}\mathbf{z}, \quad (2)$$

where $c(v_M(\mathbf{x}), \mathbf{z})$ is the similarity between $v_M(x)$ and each element in $\mathbf{z}$, namely, for every $z_m^j \in \mathbf{z}$,

$$c(v_M(\mathbf{x}), z_m^j) = L_j(\kappa)(\mathbf{x}, \mathbf{x}_m),$$

where $L_j$ is applied along the second argument of $\kappa(\cdot, \cdot)$, and $\mathbf{x}_m$ is the collocation point corresponding to $z_m^j$. Here $\mathbf{C}(\mathbf{z}, \mathbf{z})$ is the Gram matrix of $\mathbf{z}$, constructed from $Q \times Q$ sub-blocks,

$$\mathbf{C} = \begin{pmatrix} \mathbf{C}_{11} & \ldots & \mathbf{C}_{1Q} \\ \vdots & \ddots & \vdots \\ \mathbf{C}_{Q1} & \ldots & \mathbf{C}_{QQ} \end{pmatrix}, \quad (3)$$

where each $\mathbf{C}_{ij}$ is the similarity matrix associated with a pair of linear operators,

$$\mathbf{C}_{ij} = c(\mathbf{h}_i, \mathbf{h}_j) = L_i \circ L_j(\kappa)(\mathbf{h}_i, \mathbf{h}_j), \qquad (4)$$

where $1 \leq i, j \leq Q$, $L_i$ and $L_j$ are applied to the first and second arguments of $\kappa$, respectively, $\mathbf{h}_i = \{z_m^i\}_m$ and $\mathbf{h}_j = \{z_m^j\}_m$. Therefore, the size of the Gram matrix $\mathbf{C}$ is $(Q_\Omega M_\Omega + (Q - Q_\Omega)(M - M_\Omega)) \times (Q_\Omega M_\Omega + (Q - Q_\Omega)(M - M_\Omega))$.

This approach can be explained from a probabilistic perspective (Long et al., 2022). That is, we assign a GP prior over $u$, and given a sufficiently smooth kernel $\kappa$, all the linear operators over $u$, namely, $L_j(u)$ also follow a GP prior, and their projection on the collocation points $\mathcal{M}$, namely, $\mathbf{z}$, follow a multi-variate Gaussian prior distribution, $p(\mathbf{z}) \sim \mathcal{N}(\mathbf{z}|\mathbf{0}, \mathbf{C})$. Softening the outer constraints in the aforementioned nested optimization by maximizing a likelihood, this method essentially seeks for an MAP estimation of $\mathbf{z}$, and the prediction (2) is the posterior mean conditioned on $\mathbf{z}$.

## 3 Our Framework

Despite the success of (Chen et al., 2021), it requires computation of a Gram (covariance) matrix (see (3)) with dimensions typically exceeding the number of collocation points. This can exacerbate computational challenges, particularly when addressing complex PDEs that demand a considerable number of collocation points (Cho et al., 2024; Florido et al., 2024). Moreover, the construction of the Gram matrix relies on the particular set of linear operators present in the PDE, rendering it cumbersome for implementation and the adoption of efficient approximations, if needed.

We therefore propose an alternative kernel learning framework for nonlinear PDE solving, which simplifies the Gram matrix construction and computation. Specifically, we are inspired by the standard kernel/GP regression. Suppose the solution values at the collocation points are known, denoted as $\mathbf{u}_{\mathcal{M}}^* = (u^*(\mathbf{x}_1), \ldots, u^*(\mathbf{x}_M))^\top$. The optimal solution estimate within the framework of standard kernel regression takes the interpolation form: $t(\mathbf{x}) = \kappa(\mathbf{x}, \mathcal{M})\mathbf{K}_{MM}^{-1}\mathbf{u}_{\mathcal{M}}^*$, where $\mathbf{K}_{MM} = \kappa(\mathcal{M}, \mathcal{M})$ denotes the kernel matrix computed on the collocation points (of size $M \times M$). This form is derived by minimizing the RKHS norm while aligning $u^*$ at $\mathcal{M}$. In GP regression, $t(\mathbf{x})$ serves as the mean function of the posterior process.

For general PDE solving, as depicted in (1), one often lacks knowledge of the solution values at arbitrary collocation points. Therefore, we regard them as unknown, free variables denoted by $\boldsymbol{\eta}$. We model the solution estimate as

$$u(\mathbf{x}; \boldsymbol{\eta}) = \kappa(\mathbf{x}, \mathcal{M})\mathbf{K}_{MM}^{-1}\boldsymbol{\eta}. \qquad (5)$$

We then apply each linear operator $L_j$ in the PDE over $u(\mathbf{x}; \boldsymbol{\eta})$ to approximate $L_j(u^*)$. Following this way, we do not need to explicitly estimate the values of $L_j(u^*)$ at the collocation points as in (Chen et al., 2021; Long et al., 2022) (namely $z_m^j$ in $\mathbf{z}$). Correspondingly, the Gram matrix $\mathbf{K}_{MM}$ is substantially smaller (of size $M \times M$) and it is more convenient to compute — there is no need to enumerate pairs of linear operators and apply them to the kernel function to compute different sub-blocks.

The learning of our model is carried out by addressing the following constrained optimization problem:

$$\begin{cases} \underset{u \in \mathcal{U}}{\text{minimize}} \;\; \|u\|_{\mathcal{U}} \\ \text{s.t.} \; \frac{1}{M_\Omega} \sum_{m=1}^{M_\Omega} \left(\mathcal{P}(u)(\mathbf{x}_m) - f(\mathbf{x}_m)\right)^2 \\ + \frac{1}{M - M_\Omega} \sum_{m=M_\Omega+1}^{M} \left(\mathcal{B}(u)(\mathbf{x}_m) - g(\mathbf{x}_m)\right)^2 \leq \epsilon, \\ u \text{ takes the kernel interpolation form (5),} \end{cases} \qquad (6)$$

where $\epsilon \geq 0$ is a given relaxation parameter. Note that since our formulation (5) uses a reduced model space (there are no basis functions associated with operators $\{L_j\}$) as opposed to (Chen et al., 2021), we introduce $\epsilon$ to enable feasibility of the optimization and to establish the convergence; see our convergence analysis in Appendix Section A. In practice, addressing (6) directly can be cumbersome. We may opt to optimize an unconstrained objective with soft regularization instead,

$$\underset{\boldsymbol{\eta}}{\text{minimize}} \;\; \mathcal{L}(u(x; \boldsymbol{\eta}); \alpha, \beta) := \|u\|_{\mathcal{U}}^2$$
$$+ \alpha \left[ \frac{1}{M_\Omega} \sum_{m=1}^{M_\Omega} (\mathcal{P}(u)(\mathbf{x}_m) - f(\mathbf{x}_m))^2 - \epsilon/2 \right] \qquad (7)$$
$$+ \beta \left[ \frac{1}{M - M_\Omega} \sum_{m=M_\Omega+1}^{M} (\mathcal{B}(u)(\mathbf{x}_m) - g(\mathbf{x}_m))^2 - \epsilon/2 \right],$$

where $\alpha, \beta > 0$ are the regularization strengths, and $\epsilon$ can be simply set to zero.

**Efficient Computation.** In scenarios where PDEs are complex and challenging, capturing the solution details might necessitate using a vast array of collocation points. Since our framework uses the standard Gram/kernel matrix to construct the solution estimate (as illustrated in (5)), a wide range of existing kernel approximation and computation methods (Quinonero-Candela & Rasmussen, 2005; Rahimi & Recht, 2007; Farahat et al., 2011; Lindgren et al., 2011) can be readily employed to accelerate computation involving $\mathbf{K}_{MM}^{-1}$. This facilitates the reduction of computational costs and enables scalability to accommodate massive collocation points.

As an instance, we propose to induce a Kronecker product structure to accelerate computation and scale to a large number of collocation points. Specifically, we place the collocation points on a grid, namely, $\mathcal{M} = \mathbf{s}^1 \times \ldots \times \mathbf{s}^d$, where each $\mathbf{s}^k$ includes a collection of locations at input dimension

$k$, *i.e.*, $\mathbf{s}^k = (s_1^k, \ldots, s_{m_k}^k)^\top \in \mathbb{R}^{m_k}$. These locations can be regular-spaced or randomly sampled. Accordingly, $\mathcal{M}$ is an $d$-dimensional array of size $m_1 \times \ldots \times m_d$. Next, we employ a product kernel that is decomposed as along the input dimensions, $\kappa(\mathbf{x}, \mathbf{x}') = \prod_{j=1}^d \kappa_j(x_j, x_j')$, where each $\kappa_j$ is a kernel function of two scalar variables at input dimension $j$. An example is the widely-used Square Exponential (SE) kernel. The product kernel corresponds to a tensor product structure in the latent feature space, offering rich representational power to approximate PDE solutions; see the theoretical justification in (Wang et al., 2021).

With the product kernel, the kernel matrix on the collocation points $\mathcal{M}$ becomes a Kronecker product, $\mathbf{K}_{MM} = \mathbf{K}_1 \otimes \ldots \otimes \mathbf{K}_d$, where each $\mathbf{K}_j = \kappa_j(\mathbf{s}^j, \mathbf{s}^j)$ is a local kernel matrix for dimension $j$ ($1 \leq j \leq d$), of size $m_j \times m_j$. We then leverage the Kronecker product properties to efficiently compute the solution estimate (5) as

$$\begin{aligned}
&\left[\kappa_1(x_1, \mathbf{s}^1) \otimes \ldots \otimes \kappa_d(x_d, \mathbf{s}^d)\right]\left[\mathbf{K}_1 \otimes \ldots \otimes \mathbf{K}_d\right]^{-1} \boldsymbol{\eta} \\
&= \left[\kappa_1(x_1, \mathbf{s}^1)\mathbf{K}_1^{-1} \otimes \ldots \otimes \kappa_d(x_d, \mathbf{s}^d)\mathbf{K}_d^{-1}\right] \boldsymbol{\eta} \qquad (8) \\
&= \mathcal{A} \times_1 \left[\kappa_1(x_1, \mathbf{s}^1)\mathbf{K}_1^{-1}\right] \times_2 \ldots \times_d \left[\kappa_d(x_d, \mathbf{s}^d)\mathbf{K}_d^{-1}\right],
\end{aligned}$$

where $\mathcal{A}$ is the tensor view of $\boldsymbol{\eta}$, namely reshaping $\boldsymbol{\eta}$ as a $m_1 \times \ldots \times m_d$ array, and $\times_k$ is the mode-$k$ tensor-matrix multiplication (Kolda, 2006). In this way, we avoid explicitly computing the full kernel matrix $\mathbf{K}_{MM}$ and its inverse. We only need to invert each local kernel matrix $\mathbf{K}_j$, and hence the cost is substantially reduced. For example, considering a $100 \times 100 \times 100$ grid, the full kernel matrix is $10^6 \times 10^6$, rendering it computationally prohibitive and impracticable for most hardware. By using (8), we only need to invert three $100 \times 100$ local kernel matrices, which is cheap and fast. Furthermore, since the kernel is decomposed across individual dimensions, taking derivatives over the solution estimate $u$ will maintain the structure, *e.g.*, $\partial_{x_1 x_d} u(\mathbf{x}; \boldsymbol{\eta}) = \mathcal{A} \times_1 \left[\partial_{x_1}\kappa_1(x_1, \mathbf{s}^1)\mathbf{K}_1^{-1}\right] \times_2 \ldots \times_d \left[\partial_{x_d}\kappa_d(x_d, \mathbf{s}^d)\mathbf{K}_d^{-1}\right]$.

We then leverage the structure (8) to efficiently minimize the objective (6) or (7). The computation of each $\mathcal{P}(u)(\mathbf{x}_m)$ and $\mathcal{B}(u)(\mathbf{x}_m)$ is a straightforward application of the operators $\mathcal{P}$ and $\mathcal{B}$ to (8) and then evaluate them at the collocation points. This can be done by automatic differential libraries, such as JAX (Frostig et al., 2018). The RKHS norm in (6) and (7) can be efficiently computed by

$$\begin{aligned}
\|\mathbf{u}\|_{\mathcal{U}}^2 &= \boldsymbol{\eta}^\top \mathbf{K}_{MM}^{-1}\boldsymbol{\eta} = \boldsymbol{\eta}^\top \left[\mathbf{K}_1 \otimes \ldots \otimes \mathbf{K}_d\right]^{-1} \boldsymbol{\eta} \\
&= \boldsymbol{\eta}^\top \text{vec}(\mathcal{A} \times_1 \mathbf{K}_1^{-1} \times_2 \ldots \times_d \mathbf{K}_d^{-1}). \qquad (9)
\end{aligned}$$

We can apply any gradient-based optimization algorithm.

## 4 Convergence Analysis

We now show the convergence of our method. We inherit the road-map of (Batlle et al., 2023) and maintain the same

assumption about PDE stability and the regularity of the domain and boundary (Batlle et al., 2023, Assumption 3.7) [1], with a slight modification.

**Assumption 4.1.** The following conditions hold:

- *(C1) (Regularity of the domain and its boundary)* $\Omega \subset \mathbb{R}^d$ with $d > 1$ is a compact set and $\partial\Omega$ is a smooth connected Riemannian manifold of dimension $d - 1$ endowed with a geodesic distance $\rho_{\partial\omega}$.

- *(C2) (Stability of the PDE)* $\exists k, t \in \mathbb{N}$ with $k > d/2$ and $t > (d-1)/2$, and $\exists s, l \in \mathbb{R}$ such that for any $r > 0$, it holds that $\forall u_1, u_2 \in B_r(H^l(\Omega))$,

$$\begin{aligned}
\|u_1 - u_2\|_{H^l(\Omega)} \leq C\big(&\|\mathcal{P}(u_1) - \mathcal{P}(u_2)\|_{H^0(\Omega)} \\
&+ \|\mathcal{B}(u_1) - \mathcal{B}(u_2)\|_{H^0(\partial\Omega)}\big), \qquad (10)
\end{aligned}$$

and $\forall u_1, u_2 \in B_r(H^s(\Omega))$,

$$\begin{aligned}
\|\mathcal{P}(u_1) - \mathcal{P}(u_2)\|_{H^k(\Omega)} &+ \|\mathcal{B}(u_1) - \mathcal{B}(u_2)\|_{H^t(\partial\Omega)} \\
&\leq C\|u_1 - u_2\|_{H^s(\Omega)}, \qquad (11)
\end{aligned}$$

where $C = C(r) > 0$ is a constant independent of $u_1$ and $u_2$, $B(r)$ is an open ball with radius $r$, $H^j = W^{j,2}$ is a Sobolev space where each element and its weak derivatives up to the order of $j$ have a finite $L^2$ norm.

- *(C3)* The RKHS $\mathcal{U}$ is continuously embedded in $H^{s+\tau}(\Omega)$ where $\tau > 0$.

**Lemma 4.2.** *Let $u^* \in \mathcal{U}$ denote the unique strong solution of* (1). *Suppose Assumption 4.1 is satisfied, and a set of collocation points $\mathcal{M} \subset \overline{\Omega}$ is given, where $\mathcal{M}_\Omega \subset \mathcal{M}$ denotes the collocation points in the interior $\Omega$ and $\mathcal{M}_{\partial\Omega} \subset \mathcal{M}$ the collocation points on the boundary $\partial\Omega$. Assume the Voronoi diagram based on the collocation points has a uniformly bounded aspect ratio across all the cells [2]. Define the fill-distances*

$$\begin{aligned}
h_\Omega &:= \sup_{\mathbf{x} \in \Omega} \inf_{\mathbf{x}' \in \mathcal{M}_\Omega} |\mathbf{x} - \mathbf{x}'|, \\
h_{\partial\Omega} &:= \sup_{\mathbf{x} \in \partial\Omega} \inf_{\mathbf{x}' \in \mathcal{M}_{\partial\Omega}} \rho_{\partial\Omega}(\mathbf{x}, \mathbf{x}'), \qquad (12)
\end{aligned}$$

*where $|\cdot|$ is the Euclidean distance, and $\rho_{\partial\Omega}$ is a geodesic distance defined on $\partial\Omega$. Set $h = max(h_\Omega, h_{\partial\Omega})$. There is*

---

[1] Note that this assumption, along with its minor variants, is considered mild and widely used in convergence analysis. For many examples of nonlinear PDEs that satisfy this assumption, see (Batlle et al., 2023).

[2] Specifically, for each cell $\mathcal{T}_m$ that includes $\mathbf{x}_m$, the aspect ratio is defined as the ratio between the smallest radius of a ball centered at $\mathbf{x}_m$ containing the closure of $\mathcal{T}_m$ and the maximum radius of the ball centered at $\mathbf{x}_m$ contained in $\mathcal{T}_m$. The aspect ratio is assumed to be uniformly bounded across $m$ and the total number of collocation points.

*always a minimizer of* (6) *with the set of collocation points* $\mathcal{M}$ *and* $\epsilon = C_0 h^{2\tau}$ *where* $C_0 > 0$ *is a sufficiently large constant independent of* $h$. *Let* $u^\dagger$ *denote such a minimizer. When* $h$ *is sufficiently small, at least* $h \leq C_1 M^{-\frac{1}{d}}$ *where* $C_1 > 0$ *is a constant, then*

$$\|u^\dagger - u^*\|_{H^l(\Omega)} \leq C h^\rho \|u^*\|_{\mathcal{U}}, \tag{13}$$

*where* $\rho = \min(k, t, \tau)$, *and* $C > 0$ *is independent of* $u^\dagger$ *and* $h$.

**Proposition 4.3.** *Given the set of collocation points* $\mathcal{M}$ *and* $\epsilon = C_0 h^{2\tau}$ *where* $C_0 > 0$ *is a sufficiently large constant, there exists* $\alpha_M, \beta_M > 0$ *such that the minimizer of* (7) *with* $\alpha = \alpha_M$ *and* $\beta = \beta_M$ *is also the minimizer of* (6). *That means, with proper choices of the regularization strengths, the minimizer of* (7) *enjoys the same convergence result as in* (13).

We can see the convergence results of our framework are as comparably strong as the results for the method of Chen et al. (2021); see (Batlle et al., 2023, Theorem 3.8), though the latter employs a richer model space. We leave the proof in Section A and B of the Appendix.

## 5 Related Work

The prior works of Graepel (2003); Raissi et al. (2017) propose Gaussian Process (GP) models for solving linear PDEs in the presence of noisy measurements of source terms. Chen et al. (2021) introduced a kernel method capable of solving both linear and nonlinear PDEs. The solution approximation is constructed by both kernels and kernel derivatives (more generally, the linear operators of the PDE over the kernels). Hence, the differentiation operators need to be embedded into the kernels to construct the Gram matrix whose dimension is typically greater than the number of collocation points. Long et al. (2022) proposed a general framework for incorporating differential equations into GP learning, which is conceptually equivalent to (Chen et al., 2021). However, Long et al. (2022) formulated the method within a probabilistic modeling framework and developed a variational posterior inference algorithm. In (Batlle et al., 2023), a systematic theoretical framework is established to analyze the convergence of the method of Chen et al. (2021). To alleviate the computational challenge for massive collocation points, Chen et al. (2023) adapted the sparse inverse Cholesky factorization (Schafer et al., 2021) to approximate the Gram matrix of (Chen et al., 2021). An alternative approach was proposed by Meng & Yang (2023), which adjusts the Nyström method (Jin et al., 2013) to obtain a sparse approximation of the Gram matrix. Despite the success of these methods, the construction of the sparse approximation needs to carefully handle different sub-blocks in the Gram matrix, where each sub-block corresponds to a pair of linear operators over the kernels, and hence it is complex and

relatively inconvenient for implementation. In our work, the solution is approximated by a standard kernel interpolation, and the Gram matrix is therefore just the kernel matrix over the collocation points. The size of the Gram matrix is smaller. More important, the existent sparse approximation methods can be readily applied to our model, without the need for complex adjustments or novel development.

The computational efficiency of Kronecker product structures has been recognized in various works (Saatçci, 2012; Wilson & Nickisch, 2015; Izmailov et al., 2018; Zhe et al., 2019). Wilson et al. (2015) highlighted that utilizing a regular (evenly-spaced) grid results in Toeplitz-structured kernel matrices, facilitating $O(n \log n)$ computation. However, in typical machine learning applications, data is not observed on a grid, limiting the utility of the Kronecker product. In contrast, for PDE solving, estimating solution values on a grid is natural, making Kronecker products combined with kernels a promising avenue for efficient computation. The recent work (Fang et al., 2024) uses a similar computational method to solve high-frequency and multi-scale PDEs. The major contribution is to introduce a spectral mixture kernel in each dimension to capture the dominant frequencies in the kernel space. This work can be viewed as an instance of our proposed framework. We in addition give a theoretical analysis about the convergence of our framework. Extensive discussions on Bayesian learning and PDE problems are given in (Owhadi, 2015).

Our work is related to radial basis function (RBF) methods (Hardy, 1971; Kansa, 1990; Powell, 1992; Buhmann, 2000; Tolstykh & Shirobokov, 2003), which approximate the PDE solution as a linear combination of RBF basis functions, such as Multiquadrics and Gaussian: $u(\mathbf{x}) \approx \sum_j \alpha_j \psi_j(\mathbf{x})$, where $\psi_j(\mathbf{x}) \triangleq \psi(\|\mathbf{x} - \mathbf{x}_j\|)$, $\{\mathbf{x}_j\}$ denote collocation points, and $\|\cdot\|$ is usually chosen as the $L_2$ norm. To estimate the coefficients $\boldsymbol{\alpha} = \{\alpha_j\}$, RBF methods typically convert the PDE into a linear system $\mathbf{A}\boldsymbol{\alpha} = \mathbf{b}$. There are also methods that estimate the coefficients via residual minimization, *e.g.*, (Böhmer & Schaback, 2020). A key distinction between our method and RBF methods lies in the solution modeling. Our method uses a kernel interpolation form (5) to approximate the solution. We can draw an implicit connection via setting $\boldsymbol{\alpha} = \mathbf{K}_{MM}^{-1}\boldsymbol{\eta}$, where we directly estimate $\boldsymbol{\eta}$ — the solution values at the collocation points. One might question why not optimize $\boldsymbol{\alpha}$ instead. Empirically, we found that, however, doing so significantly degrades performance — often by one order of magnitude in our test benchmarks. This degradation might stem from the fact that each coefficient in $\boldsymbol{\alpha}$ has a strong global influence on the *entire* solution approximation, making the optimization process more challenging, especially for nonlinear PDEs. Additional details and ablation studies supporting this observation are provided in Appendix Section C.5. It is worth noting that while local RBF methods (Fornberg

& Flyer, 2015; Bayona et al., 2010) also approximate the solution and its derivatives via interpolation of nodal values, they primarily rely on neighborhood points through stencils and require repeatedly computing the interpolation weights across different neighborhoods. In contrast, our method employs a single kernel interpolation, offering both convenience and computational efficiency.

# 6 Numerical Experiments

To evaluate our method, we considered four commonly-used benchmark PDE families in the literature of machine learning based solvers (Raissi et al., 2019; Chen et al., 2021).

**The Burgers' Equation**. We first tested with a viscous Burgers' equation,

$$u_t + uu_x - \nu u_{xx} = 0, \quad \forall (x,t) \in (-1,1) \times (0,1],$$
$$u(x,0) = -\sin(\pi x), \quad u(-1,t) = u(1,t) = 0. \quad (14)$$

The solution is computed from the Cole–Hopf transformation with numerical quadrature (Chen et al., 2021). We considered two cases: $\nu = 0.02$, and $\nu = 0.001$.

**Nonlinear elliptic PDE.** We next tested with the instance of nonlinear elliptic PDE used in (Chen et al., 2021),

$$-\Delta u(\mathbf{x}) + u^3(\mathbf{x}) = f(\mathbf{x}), \quad \forall \mathbf{x} \in \Omega,$$
$$u(\mathbf{x}) = 0, \quad \forall \mathbf{x} \in \partial\Omega, \quad (15)$$

where $\Omega = [0,1]^2$, the solution is crafted as $u(\mathbf{x}) = \sin(\pi x_1)\sin(\pi x_2) + 4\sin(4\pi x_1)\sin(4\pi x_2)$, and $f(\mathbf{x})$ is correspondingly computed via the equation.

**Eikonal PDE.** Third, we tested with a regularized Eikonal equation as used in (Chen et al., 2021),

$$|\nabla u(\mathbf{x})|^2 = f(\mathbf{x})^2 + \epsilon\Delta u(\mathbf{x}), \quad \forall \mathbf{x} \in \Omega,$$
$$u(\mathbf{x}) = 0, \quad \forall \mathbf{x} \in \partial\Omega, \quad (16)$$

where $\Omega = [0,1]^2$, $f(\mathbf{x}) = 1$, and $\epsilon = 0.1$. The solution is computed from a highly-resolved finite difference solver as provided by (Chen et al., 2021).

**Allen-Cahn Equation**. Fourth, we considered a 2D stationary Allen-Cahn equation with a source function and Dirichlet boundary conditions.

$$u_{xx} + u_{yy} + \gamma(u^m - u) = f(x,y), \quad (17)$$

where $\gamma = 1$, $m = 3$, and $(x,y) \in [0,1]^2$. We crafted the solution in the form of $u = \sin(2\pi a x_1)\cos(2\pi a x_2) + \sin(2\pi x_1)\cos(2\pi x_2)$, and $f$ is computed through the equation. We tested with $a = 15$ and $a = 20$.

**Method and Settings.** We implemented our method with JAX (Frostig et al., 2018). We denote our

method as SKS (Simple Kernel-based Solver) We compared with (Chen et al., 2021) that uses kernel and kernel derivatives (more generally, linear operators) to approximate the solution, which we denote as DAKS (Derivative-Augmented Kernel-based Solver). We used the original implementation from the authors[3]. In addition, we compared with physics-informed neural network (PINN) (Raissi et al., 2019), a mainstream machine learning based PDE solver. The PINN is implemented with PyTorch (Paszke et al., 2019). For SKS, we minimize (7) (with $\epsilon = 0$), and used ADAM optimization with learning rate $10^{-3}$. The maximum number of epochs was set to 1M. We stopped the optimization if the loss stopped improving for 1K updates. For DAKS, we used the relaxed Gauss-Newton optimization developed in the original paper. The PINN was first trained by 10K ADAM epochs with learning rate $10^{-3}$ and then by L-BFGS with learning rate $10^{-1}$ with a maximum of 50K iterations. The tolerance level for L-BFGS was set to $10^{-9}$. To identify the architecture for the PINN, we varied the number of layers from {2, 3, 5, 8, 10}, and the width of each layer from {10, 20, 30, . . . , 100}. We used *tanh* as the activation function. For DAKS and SKS, we used Square Exponential (SE) kernel with different length-scales across the input dimensions. We selected the nugget term from {5E-5, 1E-5, 5E-6, 1E-6, . . . , 1E-13}. However, for solving the nonlinear elliptic PDE with DAKS, we used its default approach that assigns an adaptive nugget for the two sub-blocks in the Gram matrix. This gives the best performance for DAKS. The length-scales were selected from a grid search, from $[0.1, 0.2]^2$ for the nonlinear elliptic and Eikonal PDEs, $[0.05, 0.01]^2$ for Allen-Cahn equation, and $[0.003, 0.05] \times [0.02, 0.3]$ for Burgers' equation. For SKS, we selected $\alpha$ and $\beta$ in (7) from the range $\{10^{-2}, 10^{-1}, \ldots, 10^{10}, 10^{12}, 10^{14}, 10^{15}, 10^{20}\}$, jointly with other hyperparameters, including the kernel length scales and nugget terms. To efficiently tune the hyperparameters, we first performed a random search to identify a promising group of hyperparameters. We then fixed all other hyperparameters and conducted a grid search over $\alpha$ and $\beta$. Finally, we fixed $\alpha$ and $\beta$ and performed a grid search over the remaining hyperparameters. We reported the best solution error of each method throughout the running. In Section C.4 of Appendix, we further examined the sensitivity of our method to the kernel parameters.

## 6.1 Solution Accuracy

**Simpler Cases.** We first tested all the methods on less challenging benchmarks, for which a small number of collocation points is sufficient. Specifically, we tested with Burgers' equation with viscosity $\nu = 0.02$, the nonlinear elliptic PDE, and Eikonal PDE. These are the same test cases employed

---

[3]https://github.com/yifanc96/
NonLinPDEs-GPsolver

in (Chen et al., 2021). Following (Chen et al., 2021), we varied the number of collocation points from {600, 1200, 2400, 4800} for Burgers' equation, and {300, 600, 1200, 2400} for nonlinear elliptic PDE and Eikonal PDE. In (Chen et al., 2021), the collocation points are randomly sampled, and hence we used the average $L^2$ of DAKS from ten runs on different sets of randomly sampled collocation points for comparison. For SKS, we used a regularly-spaced, square grid, for which the total number of grid points is close to that used for DAKS. Note that the size of the Gram matrix of DAKS is larger than SKS. We also examined running DAKS with the same set of grid points as used by SKS, but the performance is worse than randomly selected collocation points. See Appendix Section C.6 for details. We ran PINN on the same set of grid points used by SKS. The $L^2$ error is reported in Table 1. It can be seen that in most cases, SKS achieves smaller solution error than DAKS, with the exception for Burgers[4] and nonlinear elliptic PDE using 4800 and 2400 collocation points, respectively. This might be because DAKS needs to explicitly estimate more variables, including all kinds of linear operators (*e.g.*, derivatives) over the solution at the collocation points, which increases the optimization workload. In addition, the influence of the nugget term might vary across different heterogeneous blocks in the Gram matrix, which complicates the optimization. In most cases, the PINN shows worse performance than SKS, except on Burgers' equation with 600, 1200 and 2400 collocation points. The results have shown that in regimes where the computation of the Gram matrix is not a bottleneck, our method SKS, though adopting a simpler model design, can still achieve comparable or even better solution accuracy.

**Difficult Cases.** Next, we tested with more challenging cases, for which massive collocation points are necessary. These cases include Burgers' equation with $\nu = 0.001$, and Allen-Cahn equation with $a = 15$ and $a = 20$. For Burgers' equation, we found empirically that the spatial resolution is more important than the time resolution, so we set the ratio between the spatial and time resolutions to 3:1. For Allen-Cahn, we still used a square-shaped grid. We provide a more detailed ablation study in Appendix Section C.2. To verify the necessity of using massive collocation points, we first ran all the methods with the same number of collocation points as adopted in the simpler PDEs, namely, a few hundreds and/or thousands. As we can see from Table 2, the solution errors of all the methods are large, typically around the level of $10^{-1}$, indicating failures. Note that, however, when the number of collocation points increases to 4800, SKS can achieve an $L^2$ error at level $10^{-4}$ for solving the 2D Allen-Cahn equation, while the other methods still struggle

---

[4]However, if we use a $96 \times 50$ grid (still including 4800 collocation points), SKS gives $L^2$ error 7.54E-05, which can surpass DAKS. See the ablation study in Appendix Section C.2 for more details.

Table 1: $L^2$ error of solving less challenging PDEs, with a small number of collocation points. Inside the parenthesis of each top row indicates the grid used by SKS, which takes approximately the same number of collocation used by DAKS. Note that the Gram matrix of DAKS is larger than SKS.

(a) The Burgers' equation (14) with viscosity $\nu = 0.02$

| Method | 600 $(25 \times 25)$ | 1200 $(35 \times 35)$ | 2400 $(49 \times 49)$ | 4800 $(70 \times 70)$ |
|---|---|---|---|---|
| DAKS | 1.75E-02 | 7.90E-03 | 8.65E-04 | **9.76E-05** |
| PINN | **2.68E-03** | **6.72E-04** | **3.60E-04** | 3.73E-04 |
| SKS | 1.44E-02 | 5.40E-03 | 7.83E-04 | 3.21E-04 |

(b) Nonlinear elliptic PDE (15)

| Method | 300 $(18 \times 18)$ | 600 $(25 \times 25)$ | 1200 $(35 \times 35)$ | 2400 $(49 \times 49)$ |
|---|---|---|---|---|
| DAKS | 1.15E-01 | 1.15E-04 | 8.65E-04 | **1.68E-07** |
| PINN | 3.39E-01 | 1.93E-02 | 1.28E-03 | 3.20E-04 |
| SKS | **1.26E-02** | **6.93E-05** | **6.80E-06** | 1.83E-06 |

(c) Eikonal PDE (16)

| Method | 300 | 600 | 1200 | 2400 |
|---|---|---|---|---|
| DAKS | 1.01E-01 | 1.64E-02 | 2.27E-04 | 7.78E-05 |
| PINN | 2.95E-02 | 1.26E-02 | 4.53E-03 | 3.50E-03 |
| SKS | **6.23E-04** | **2.68E-04** | **1.91E-04** | **2.51E-05** |

at $10^{-1}$ error level or even bigger. Together this indicates that a much larger number of collocation points is needed.

We then ran SKS and PINN with greatly increased collocation points, *i.e.,* dense grids, varying from 6400 to 120K. In such scenarios, running DAKS becomes extremely costly or even infeasible[5]. We therefore only report the results of SKS and PINN, as shown in Table 3. One can see that the solution error of SKS is substantially reduced, achieving $10^{-3}$ for Burgers' and $10^{-4}$ to $10^{-6}$ for Allen-Cahn. It is worth noting that PINN using the same set of collocation points also arrives at the $10^{-3}$ level $L^2$ error for Burgers' but the error on Allen-Cahn is still very large, with nearly no improvement upon using much fewer collocation points. This might be due to that the relatively high frequencies in the solution (see (17)) are difficult to be captured by neural networks, due to their known "spectral bias" (Rahaman et al., 2019). Thanks to our model design (2), we can induce a Kronecker product structure in the Gram matrix to scale to massive collocation points, without the need for designing complex approximations. Note that if a product kernel is used for DAKS, each block $\mathbf{C}_{ij}$ in (3) will form a Kronecker product as well. However, the entire Gram matrix will not exhibit this structure. Consequently, the efficient computation method of SKS cannot be applied to DAKS. To directly examine the influence of the fill distance $h$ (*i.e.,* mesh norm), Appendix Fig. 1a and Fig. 1b show how the

---

[5]For 6400 collocation points, the size of the Gram matrix of DAKS is $19200 \times 19200$ for Burgers' and 2D Allen-Cahn, because there are three linear operators in each equation.

Table 2: $L^2$ error of solving more challenging PDEs with a small number of collocation points. The grids used in SKS are the same as in Table 1a.

(a) The Burgers' equation (14) with viscosity $\nu = 0.001$

| Method | 600 | 1200 | 2400 | 4800 |
|--------|-----|------|------|------|
| DAKS | 3.87E-01 | 3.12E-01 | 3.60E-01 | 2.37E-01 |
| PINN | **1.27E-01** | 2.59E-01 | 3.18E-01 | 2.65E-01 |
| SKS | 1.34E-01 | **1.11E-01** | **8.04E-02** | **1.89E-02** |

(b) The 2D Allen-Cahn equation (17) with $a = 15$

| Method | 600 | 1200 | 2400 | 4800 |
|--------|-----|------|------|------|
| DAKS | 6.84E-01 | 6.62E-01 | 6.28E-01 | 5.74E-01 |
| PINN | 4.02E0 | 6.20E0 | 4.26E0 | 5.39E0 |
| SKS | **6.80E-01** | **2.1E-01** | **5.15E-03** | **9.20E-05** |

(c) The 2D Allen-Cahn equation (17) with $a = 20$

| Method | 600 | 1200 | 2400 | 4800 |
|--------|-----|------|------|------|
| DAKS | 6.81E-01 | 6.57E-01 | 6.19E-01 | 5.64E-01 |
| PINN | 4.98E0 | 5.78E0 | 5.87E0 | 3.04E0 |
| SKS | 7.07E-01 | 6.91E-01 | **1.81E-01** | **9.83E-04** |

Table 3: $L^2$ error of solving more challenging PDEs with a large number of collocation points.

(a) The Burgers' equation (14) with viscosity $\nu = 0.001$.

| Method | 43200 (360×120) | 67500 (450×150) | 97200 (540×180) | 120000 (600×200) |
|--------|-----|------|------|------|
| PINN | 4.05E-03 | 6.01E-03 | 3.94E-03 | 4.13E-03 |
| SKS | **3.90E-03** | **3.50E-03** | **2.60E-03** | **2.28E-03** |

(b) The 2D Allen-Cahn equation (17) with $a = 15.0$

| Method | 6400 (80×80) | 8100 (90×90) | 22500 (150×150) | 40000 (200×200) |
|--------|-----|------|------|------|
| PINN | 5.03E0 | 5.30E0 | 4.21E0 | 5.86E0 |
| SKS | **8.27E-05** | **3.41E-05** | **4.34E-06** | **4.44E-06** |

(c) The 2D Allen-Cahn equation (17) with $a = 20.0$

| Method | 6400 | 8100 | 22500 | 40000 |
|--------|------|------|-------|-------|
| PINN | 4.18E0 | 4.45E0 | 5.86E0 | 5.93E0 |
| SKS | **3.98E-04** | **1.82E-04** | **4.00E-05** | **2.98E-05** |

Table 4: $L^2$ Error of a finite difference solver and SKS according to the ground-truth solution.

(a) Nonlinear Elliptic PDE (15).

| Method | $18 \times 18$ | $25 \times 25$ | $35 \times 35$ | $49 \times 49$ |
|--------|------|------|------|------|
| FD | 3.36E-02 | 1.78E-02 | 9.25E-03 | 4.78E-03 |
| SKS | **1.26E-02** | **6.93E-05** | **6.80E-06** | **1.83E-06** |

(b) The 2D Allen-Cahn equation (17) with $a = 15$.

| Method | $80 \times 80$ | $90 \times 90$ | $150 \times 150$ | $200 \times 200$ |
|--------|------|------|------|------|
| FD | 8.57E-02 | 6.68E-02 | 2.33E-02 | 1.30E-02 |
| SKS | **8.27E-05** | **3.41E-05** | **4.34E-06** | **4.44E-06** |

(c) The 2D Allen-Cahn equation (17) with $a = 20$.

| Method | $80 \times 80$ | $90 \times 90$ | $150 \times 150$ | $200 \times 200$ |
|--------|------|------|------|------|
| FD | 1.62E-01 | 1.24E-01 | 4.22E-02 | 2.34E-02 |
| SKS | **3.98E-04** | **1.82E-04** | **4.00E-05** | **2.98E-05** |

error of SKS varies with $h$ — determined by the collocation points — when solving Burgers' equation with $\nu = 0.02$ and $\nu = 0.001$, respectively.

**Point-wise Error.** For a fine-grained comparison, we showcase the point-wise error of each method in solving Burgers' ($\nu = 0.001$) and 2D Allen-Cahn equations. The results and discussion are given by Appendix Section C.1.

**Ablation Study on Grid Shape.** We further examined the influence of the grid shape on the solution accuracy. We compared different choices on Burgers' equation with $\nu = 0.02$ and $\nu = 0.001$. We leave the details in Appendix Section C.2.

**Comparison with Conventional Numerical Methods.** In addition to comparing SKS with ML-based solvers, we also compared it to a finite difference solver — a widely used conventional numerical approach. We discretized the PDE using numerical differences, specifically employing a centered second-order numerical difference to approximate the derivatives. The equation was then solved using a Newton-Krylov solver, which computes the inverse of the Jacobian through an iterative Krylov method. We tested on solving the nonlinear elliptic PDE (15) and the Allen-cahn equation (17), since the ground-truth solutions of these PDEs are known and we can conduct a fair comparison. The results are presented in Table 4. Note that the $L^2$ errors of PINN and SKS have already been reported in Table 1b, 3b and 3c. Our method (SKS) consistently outperforms finite difference. In most cases, the error of SKS is several orders of magnitudes smaller. It implies that using the same grid, SKS is much more efficient in approximating the solution. In addition, with the growth of the grid size, the relative

improvement of our method is often more significant, in particular when solving the nonlinear elliptic PDE with the grid $18 \times 18$ increasing to $25 \times 25$, and Allen-Cahn ($a = 15$) with the grid $90 \times 90$ increasing to $150 \times 150$. Since SKS is efficient in handling a large number of collocation points (*i.e.*, dense grid), it shows the potential of SKS.

**Irregular-shaped Domains.** While our efficient computation is performed on grids, our method can be applied to irregular-shaped domains by introducing a (minimal) virtual grid that encompasses such domains. This allows SKS to be used without any modifications. To validate the effectiveness of this strategy, we conducted additional tests, solving the nonlinear elliptic PDE (15) on a circular domain and the Allen-Cahn equation (17) on a triangular domain. In both cases, our method achieved reasonably good accuracy. Detailed results and discussions are provided in Appendix Section C.3.

**Running Time.** Finally, we examined the wall-clock runtime of SKS. We analyzed the runtime of each method when solving Burgers' equation $\nu = 0.001$ and the Allen-Cahn equation ($a = 15$) with varying numbers of collocation points. We ran the experiment on a Linux workstation equipped with an Intel(R) Xeon(R) Platinum 8360H Processor with 24GB memory. The results are presented in Appendix Table 9. SKS is several orders of magnitude faster than both DAKS and PINN per iteration. However, since DAKS employs the Gauss-Newton method, it converges much faster than the ADAM optimizer used by SKS and PINN. Nonetheless, the overall runtime of SKS is still less than 25% of that of DAKS when solving Burger's equation, and is close to DAKS when solving Allen-Cahn equation. Additionally, SKS can handle a much larger number of collocation points than DAKS. Overall, the runtime of SKS is significantly less than that of PINN.

In addition, we examined the runtime of our model using the naive full matrix computation (*i.e.,* without exploiting the Kronecker product structure). As shown, the per-iteration runtime with naive matrix operations is consistently around 100x slower compared to our method leveraging Kronecker product properties. Furthermore, when the number of collocation points increases to 22,500 for the Burgers' equation and 43,200 for the Allen-Cahn equation, the naive approach exceeds available memory, resulting in out-of-memory errors — rendering it infeasible to run.

## 7   Conclusion

We have proposed a new kernel method for nonlinear PDE solving. We use a standard kernel interpolation to model the solution estimate, which allows more convenient implementation and efficient computation. Our method can easily scale to massive collocation points, which can be important for solving challenging PDEs. The performance on a series of benchmarks is encouraging. In the future, we plan to develop more efficient optimization, *e.g.*, Gaussian-Newton, to further accelerate our method.

## Acknowledgments

HO and SZ acknowledge support from the Air Force Office of Scientific Research under MURI award number FA9550-20-1-0358 (Machine Learning and Physics-Based Modeling and Simulation). HO acknowledges support from the Air Force Office of Scientific Research under MURI award number FOA-AFRL-AFOSR-2023-0004 (Mathematics of Digital Twins), the Department of Energy under award number DE-SC0023163 (SEA-CROGS: Scalable, Efficient, and Accelerated Causal Reasoning Operators, Graphs and Spikes for Earth and Embedded Systems), the National Science Foundations under award number 2425909 (Discovering the Law of Stress Transfer and Earthquake Dynamics in a Fault Network using a Computational Graph Discovery Approach), and from the DoD Vannevar Bush Faculty Fellowship Program. SZ acknolwedges support from NSF CAREER Award IIS-2046295, and NSF OAC-2311685.

SZ also thanks Bamdad Hosseini for insightful discussions.

## Impact Statement

This paper presents work whose goal is to advance Machine Learning for PDE solving. There are many potential societal consequences of our work, none of which we feel must be specifically highlighted here.

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

# Appendix

## A   Proof of Lemma 4.2

Since for any collocation point $\mathbf{x}_m$, $\mathcal{P}(u^*)(\mathbf{x}_m) = f(\mathbf{x}_m)$ when $\mathbf{x}_m \in \Omega$ and $\mathcal{B}(u^*)(\mathbf{x}_m) = g(\mathbf{x}_m)$ when $\mathbf{x}_m \in \partial\Omega$, we can re-write (6) as

$$\begin{cases} \underset{u \in \mathcal{U}}{\text{minimize}} \ \ \|u\|_{\mathcal{U}} \\ \text{s.t.} \ \frac{1}{M_\Omega} \sum_{m=1}^{M_\Omega} \left(\mathcal{P}(u)(\mathbf{x}_m) - \mathcal{P}(u^*)(\mathbf{x}_m)\right)^2 \\ + \frac{1}{M - M_\Omega} \sum_{m=M_\Omega+1}^{M} \left(\mathcal{B}(u)(\mathbf{x}_m) - \mathcal{B}(u^*)(\mathbf{x}_m)\right)^2 \leq \epsilon, \\ u \text{ takes the kernel interpolation form (5).} \end{cases} \tag{18}$$

**Step 1.** In the first step, we show that for all $\epsilon$ above some threshold (depending on $h$), there exists a minimizer $u^\dagger$ for (18), and we would like also to bound the RKHS norm of $u^\dagger$, namely $\|u^\dagger\|_{\mathcal{U}}$. To this end, we utilize an intermediate optimization problem,

$$\begin{cases} \underset{u \in \mathcal{U}}{\text{minimize}} \ \ \|u\|_{\mathcal{U}} \\ \text{s.t.} \ u(\mathbf{x}_m) = u^*(\mathbf{x}_m), \ \ 1 \leq m \leq M. \end{cases} \tag{19}$$

Denote the minimizer of (19) by $u_M^*$. This is a standard kernel regression problem. According to the representation theorem, $u_M^*$ takes the kernel interpolation form (5), and $\|u_M^*\|_{\mathcal{U}} \leq \|u^*\|_{\mathcal{U}}$.

Since $u_M^* - u^*$ is zero at all the collocations points in $\Omega$, according to the sampling inequality (see Proposition A.1 of (Batlle et al., 2023)), when the fill-distance $h_\Omega$ is sufficiently small (note that $h_\Omega \leq h$),

$$\|u_M^* - u^*\|_{H^s(\Omega)} \lesssim h^\tau \|u_M^* - u^*\|_{H^{s+\tau}(\Omega)}, \tag{20}$$

where $\lesssim$ means the inequality holds with a positive constant factor multiplied by the right-hand side, and the constant is independent of the terms on both sides. Combining with (C2) of Assumption 4.1, we can obtain

$$\|\mathcal{P}(u_M^*) - \mathcal{P}(u^*)\|_{H^k(\Omega)} + \|\mathcal{B}(u_M^*) - \mathcal{B}(u^*)\|_{H^t(\partial\Omega)} \lesssim h^\tau \|u_M^* - u^*\|_{H^{s+\tau}(\Omega)}. \tag{21}$$

Since $\mathcal{U}$ is continuously embedded in $H^{s+\tau}(\Omega)$ — (C3) of Assumption 4.1, we have

$$\|u_M - u^*\|_{H^{s+\tau}(\Omega)} \lesssim \|u_M^* - u^*\|_{\mathcal{U}}. \tag{22}$$

Combining (21), (22) and the fact $\|u_M^*\|_{\mathcal{U}} \leq \|u^*\|_{\mathcal{U}}$, we have

$$\|\mathcal{P}(u_M^*) - \mathcal{P}(u^*)\|_{H^k(\Omega)} + \|\mathcal{B}(u_M^*) - \mathcal{B}(u^*)\|_{H^t(\partial\Omega)} \lesssim h^\tau \|u^*\|_{\mathcal{U}}. \tag{23}$$

According to (C2) of Assumption 4.1, since $k > \frac{d}{2}$ and $t > \frac{d-1}{2}$ , according to Sobolev embedding theorem (Adams & Fournier, 2003, Theorem 4.12), both $H^k(\Omega)$ and $H^t(\partial\Omega)$ are continuously embedded into $C^0(\Omega)$ and $C^0(\partial\Omega)$, respectively. Therefore,

$$\|\mathcal{P}(u_M^*) - \mathcal{P}(u^*)\|_{C^0(\Omega)} \lesssim \|\mathcal{P}(u_M^*) - \mathcal{P}(u^*)\|_{H^k(\Omega)},$$
$$\|\mathcal{B}(u_M^*) - \mathcal{B}(u^*)\|_{C^0(\partial\Omega)} \lesssim \|\mathcal{B}(u_M^*) - \mathcal{B}(u^*)\|_{H^t(\partial\Omega)}. \tag{24}$$

At any collocation point, we obviously have

$$\left(\mathcal{P}(u_M^*)(\mathbf{x}_m) - \mathcal{P}(u^*)(\mathbf{x}_m)\right)^2 \leq \|\mathcal{P}(u_M^*) - \mathcal{P}(u^*)\|_{C^0(\Omega)}^2,$$
$$\left(\mathcal{B}(u_M^*)(\mathbf{x}_m) - \mathcal{B}(u^*)(\mathbf{x}_m)\right)^2 \leq \|\mathcal{B}(u_M^*) - \mathcal{B}(u^*)\|_{C^0(\partial\Omega)}^2. \tag{25}$$

Combining (23), (24) and (25), we can obtain that

$$\frac{1}{M_\Omega} \sum_{m=1}^{M_\Omega} \left(\mathcal{P}(u_M^*)(\mathbf{x}_m) - \mathcal{P}(u^*)(\mathbf{x}_m)\right)^2$$
$$+ \frac{1}{M - M_\Omega} \sum_{m=M_\Omega+1}^{M} \left(\mathcal{B}(u_M^*)(\mathbf{x}_m) - \mathcal{B}(u^*)(\mathbf{x}_m)\right)^2 \leq C h^{2\tau} \|u^*\|_{\mathcal{U}}^2, \tag{26}$$

where $C > 0$ is a constant independent of $h$ and other terms in the inequality.

The result (26) means that given the collocation points $\mathcal{M}$ and $\epsilon = Ch^{2\tau}\|u^*\|^2_{\mathcal{U}}$, the feasible region of the optimization problem (18) is nonempty and at least includes $u^*_M$. Therefore, the minimizer of (18) must exist and satisfy

$$\|u^\dagger\|_{\mathcal{U}} \leq \|u^*_M\|_{\mathcal{U}} \leq \|u^*\|_{\mathcal{U}}. \tag{27}$$

**Step 2.** Next, we analyzed the error of $\mathcal{P}(u^\dagger)$ and $\mathcal{B}(u^\dagger)$. For notation convenience, we define two error functions,

$$\begin{aligned}
\xi_P(\mathbf{x}) &= \mathcal{P}(u^\dagger)(\mathbf{x}) - \mathcal{P}(u^*)(\mathbf{x}), \quad \mathbf{x} \in \Omega, \\
\xi_B(\mathbf{x}) &= \mathcal{B}(u^\dagger)(\mathbf{x}) - \mathcal{B}(u^*)(\mathbf{x}), \quad \mathbf{x} \in \partial\Omega.
\end{aligned} \tag{28}$$

We would like to bound the $L^2$ norm of the error functions, namely, $\|\xi_P\|_{H^0(\Omega)}$ and $\|\xi_B\|_{H^0(\partial\Omega)}$. We first consider the case for $\xi_P$. The idea is to decompose $\Omega$ into a Voronoi diagram based on the collocation points, resulting in $M_\Omega$ regular non-overlapping regions, $\mathcal{T}_1 \cup \ldots \cup \mathcal{T}_{M_\Omega} = \Omega$, such that each region $\mathcal{T}_i$ only includes one collocation point $\mathbf{x}_i$, and its filled-distance $h_i \lesssim h$ ($1 \leq i \leq M_\Omega$). We therefore can decompose the squared $L^2$ norm as

$$\|\xi_P\|^2_{H^0(\Omega)} = \sum_{i=1}^{M_\Omega} \int_{\mathcal{T}_i} \xi_P(\mathbf{x})^2 \mathrm{d}\mathbf{x} = \sum_{i=1}^{M_\Omega} \|\xi_P\|^2_{H^0(\mathcal{T}_i)}. \tag{29}$$

Since according to the mean inequality,

$$\xi_P(\mathbf{x})^2 = (\xi_P(\mathbf{x}) - \xi_P(\mathbf{x}_i) + \xi_P(\mathbf{x}_i))^2 \leq 2(\xi_P(\mathbf{x}) - \xi_P(\mathbf{x}_i))^2 + 2\xi_P(\mathbf{x}_i)^2,$$

we immediately obtain

$$\|\xi_P\|^2_{H^0(\mathcal{T}_i)} \lesssim \|\xi_P - \xi_P(\mathbf{x}_i)\|^2_{H^0(\mathcal{T}_i)} + \lambda(\mathcal{T}_i)\xi_P(\mathbf{x}_i)^2, \tag{30}$$

where $\lambda(\mathcal{T}_i)$ is the volume of $\mathcal{T}_i$.

The function $\xi_P - \xi_P(\mathbf{x}_i)$ takes zero at $\mathbf{x}_i$. Since the aspect ratio of $\mathcal{T}_i$ is bounded, we can apply the sampling inequality (which is also called Poincaré inequality in this case),

$$\|\xi_P - \xi_P(\mathbf{x}_i)\|_{H^0(\mathcal{T}_i)} \leq Ch_i^k\|\xi_P - \xi_P(\mathbf{x}_i)\|_{H^k(\mathcal{T}_i)} \lesssim h^k\|\xi_P - \xi_P(\mathbf{x}_i)\|_{H^k(\mathcal{T}_i)}, \tag{31}$$

where $C > 0$ is a constant depending on the aspect ratio of $\mathcal{T}_i$. Using the mean inequality again,

$$\|\xi_P - \xi_P(\mathbf{x}_i)\|^2_{H^0(\mathcal{T}_i)} \lesssim h^{2k}\left(\|\xi_P\|^2_{H^k(\mathcal{T}_i)} + \|\xi_P(\mathbf{x}_i)\|^2_{H^k(\mathcal{T}_i)}\right) = h^{2k}\left(\|\xi_P\|^2_{H^k(\mathcal{T}_i)} + \lambda(\mathcal{T}_i)\xi_P(\mathbf{x}_i)^2\right). \tag{32}$$

Since $\lambda(\mathcal{T}_i) \lesssim h^d$, combining (29), (30) and (32), we can obtain

$$\begin{aligned}
\|\xi_P\|^2_{H^0(\Omega)} &\lesssim h^{2k}\sum_i \|\xi_P\|^2_{H^k(\mathcal{T}_i)} + (h^d + h^{2k+d})\sum_i \xi_P(\mathbf{x}_i)^2 \\
&\lesssim h^{2k}\|\xi_P\|^2_{H^k(\Omega)} + (h^d + h^{2k+d}) \cdot M_\Omega \cdot \epsilon,
\end{aligned} \tag{33}$$

where $\epsilon$ comes from the constraint of (18). To ensure feasibility and to establish convergence, we set $\epsilon = Ch^{2\tau}\|u^*\|^2_{\mathcal{U}}$ as shown in (26). When $h \lesssim M^{-\frac{1}{d}}$ and is sufficiently small, we have $(h^d + h^{2k+d})M_\Omega \leq (h^d + h^{2k+d})M \leq 1 + h^{2k} \leq 2$. Therefore, we can extend the R.H.S of (33) to

$$\|\xi_P\|^2_{H^0(\Omega)} \lesssim h^{2k}\|\xi_P\|^2_{H^k(\Omega)} + h^{2\tau}\|u^*\|^2_{\mathcal{U}}. \tag{34}$$

We can follow a similar approach to show that

$$\|\xi_B\|^2_{H^0(\partial\Omega)} \lesssim h^{2t}\|\xi_B\|^2_{H^t(\partial\Omega)} + h^{2\tau}\|u^*\|^2_{\mathcal{U}}. \tag{35}$$

Combining (34) and (35),

$$\left(\|\xi_P\|_{H^0(\Omega)} + \|\xi_B\|_{H^0(\partial\Omega)}\right)^2 \lesssim h^{2\cdot\min(t,k)}\left(\|\xi_P\|_{H^k(\Omega)} + \|\xi_B\|_{H^t(\partial\Omega)}\right)^2 + h^{2\tau}\|u^*\|^2_{\mathcal{U}}. \tag{36}$$

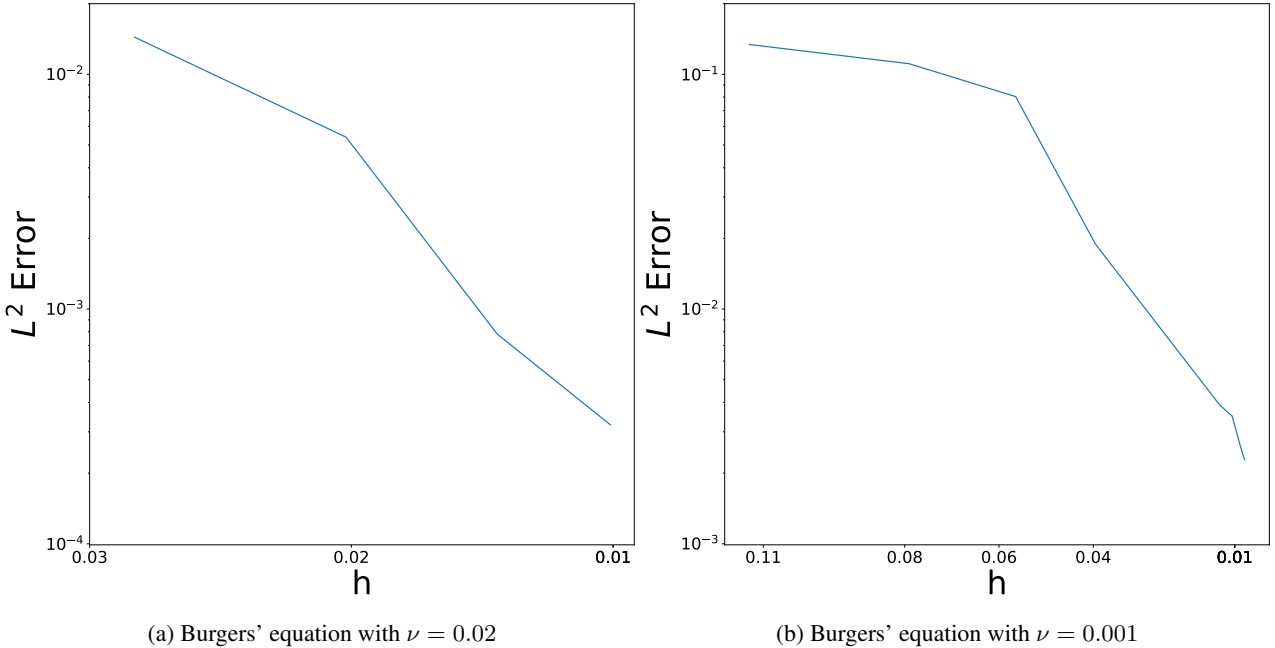

(a) Burgers' equation with $\nu = 0.02$  (b) Burgers' equation with $\nu = 0.001$

Figure 1: Fill-distance (mesh norm) $h$ *vs.* $L^2$ error of SKS.

Since $\mathcal{U} \hookrightarrow H^{s+\tau}$ — (C3) of Assumption 4.1, we have $\mathcal{U} \hookrightarrow H^s$. Leveraging (27) and (C2) of Assumption 4.1, we immediately obtain

$$\|\xi_P\|_{H^k(\Omega)} + \|\xi_B\|_{H^t(\partial\Omega)} \lesssim \|u^\dagger - u^*\|_{H^s(\Omega)} \lesssim \|u^\dagger - u^*\|_{\mathcal{U}} \lesssim \|u^*\|_{\mathcal{U}}. \tag{37}$$

Combining (36) and (37), and (C1) of Assumption 4.1, we arrive at

$$\|u^\dagger - u^*\|_{H^l(\Omega)} \lesssim h^\rho \|u^*\|_{\mathcal{U}} \tag{38}$$

where $\rho = \min(k, t, \tau)$. When $h \to 0$, $u^\dagger$ converges to $u^*$.

## B  Proof of Proposition 4.3

The constraint optimization problem (6) is equivalent to the following mini-max optimization problem,

$$\min_u \max_{w \geq 0} \|u\|_{\mathcal{U}} + w \left[ \frac{1}{M_\Omega} \sum_{m=1}^{M_\Omega} \left(\mathcal{P}(u)(\mathbf{x}_m) - \mathcal{P}(u^*)(\mathbf{x}_m)\right)^2 \right.$$
$$\left. + \frac{1}{M - M_\Omega} \sum_{m=M_\Omega+1}^{M} \left(\mathcal{B}(u)(\mathbf{x}_m) - \mathcal{B}(u^*)(\mathbf{x}_m)\right)^2 - \epsilon \right]. \tag{39}$$

Suppose the feasible region is non-empty. Denote the optimum of (39) by $(u^\dagger, w^\dagger)$. Then $u^\dagger$ is a minimizer of (6). Now if we set $\alpha = \beta = w^\dagger$ in (7), and optimizing (7) will recover the minimizer $u^\dagger$.

## C  More Results

### C.1  Point-wise Error

For a fine-grained evaluation, we examined how the point-wise error of DAKS and SKS varies along with the increase of collocation points. To this end, we altered the number of collocation points from 600, 4800 and 120K on Burgers' equation with $\nu = 0.001$, and from 600, 2400, and 40K on 2D Allen-Cahn equation with both $a = 15$ and $a = 20$. The results

are shown in Fig. 2, 3 and 4. It can be seen that across all the three PDEs, the solution error of SKS decreases more and more along with the increase of collocation points. Note that for Allen-Cahn with $a = 15$, the visual difference between SKS using 2400 and 40K collocation points is little, though numerically the difference is at three orders of magnitudes (5.15E-03 *vs.* 4.44E-06). For DAKS, the point-wise error decreases substantially as the number of collocation points grows when solving Burgers' equation (see Fig. 2), but not obviously on solving Allen-Cahn equation (see Fig. 3 and 4). This is consistent with the global error shown in Table 2. This might be because the quantities of collocation point used are not sufficient to lead to a qualitative boost of DAKS. However, scaling up to much more collocation points, such as 400K, incurs a substantial increase of the computational cost.

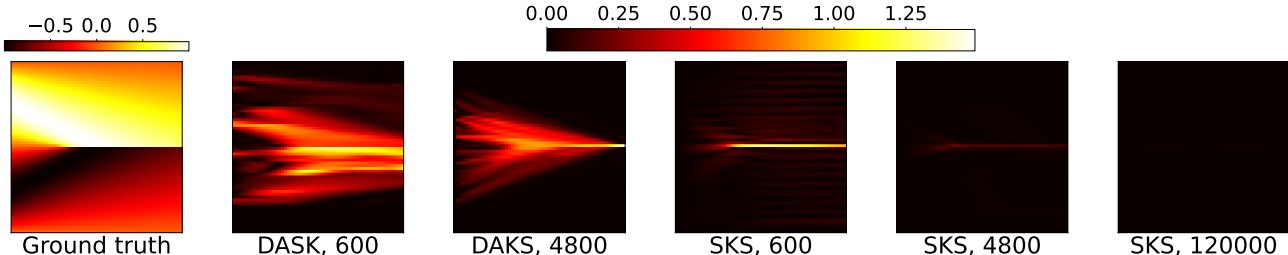

Figure 2: Point-wise solution error for Burgers' equation (14) with viscosity $\nu = 0.001$.

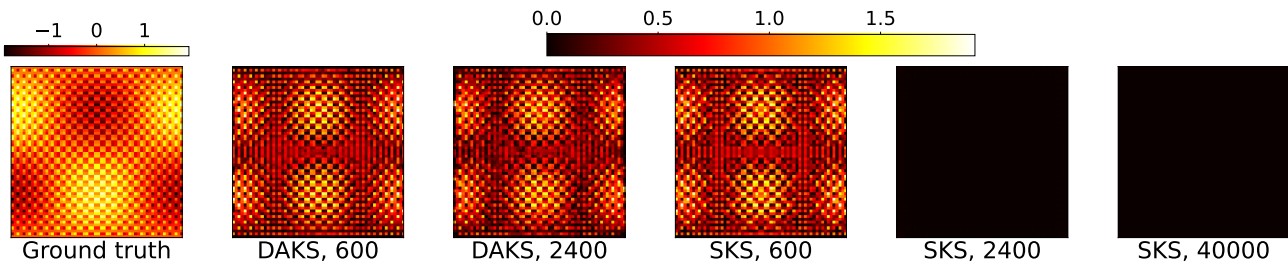

Figure 3: Point-wise solution error for 2D Allen-Cahn equation (17) with $a = 15.0$.

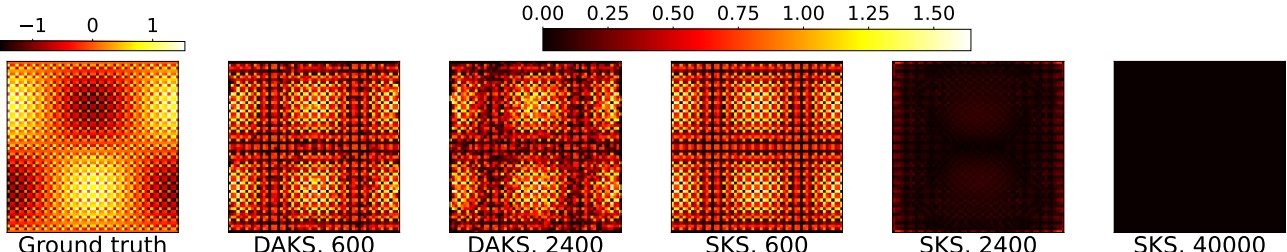

Figure 4: Point-wise solution error for 2D Allen-Cahn equation (17) with $a = 20.0$.

## C.2 Ablation Study on Grid Shape

We investigated how the grid shape can influence the performance of our method. To this end, we tested on Burgers' equation with $\nu = 0.02$ and $\nu = 0.001$. For the former case, we fixed the number of collocation points to be 4800 and varied the time resolution from 10 to 80, and the spatial resolution is obtained by dividing 4800 by the time resolution and rounding up to an integer. Similarly, for $\nu = 0.001$, we fixed the number of the collocation points to 120K, and varied the time resolution from 100 to 600. We show the $L^2$ error of using each grid shape in Table 5. It can be seen that the grid shape does influence the error. In particular, when $\nu$ is small, *i.e.,* $\nu = 0.001$, the higher the spatial resolution, the smaller the error. The smallest error is achieved when we use the space-time resolution $1200 \times 100$. The time resolution seems to have much less effect on the solution accuracy. This is reasonable, because on Burgers' equation, a smaller viscosity ($\nu$) increases the sharpness of the shock wave (spatial function). Naturally, the higher the spatial resolution, the more accurate the sharpness can be captured.

Table 5: $L^2$ error of SKS using different grid shapes to solve Burgers' equation (14). The grid shape is depicted as "spatial-resolution $\times$ time-resolution".

(a) Viscosity $\nu = 0.02$, with 4800 collocation points.

| Grid shape | $60 \times 80$ | $69 \times 70$ | $96 \times 50$ | $160 \times 30$ | $480 \times 10$ |
|---|---|---|---|---|---|
| $L^2$ error | 2.27E-03 | 4.10E-04 | **7.54E-05** | 1.72E-04 | 5.98E-04 |

(b) Viscosity $\nu = 0.001$ with 120000 collocation points.

| Grid shape | $200 \times 600$ | $240 \times 500$ | $400 \times 300$ | $600 \times 200$ | $1200 \times 100$ |
|---|---|---|---|---|---|
| $L^2$ error | 1.93E-02 | 6.68E-03 | 3.44E-03 | 2.28E-03 | **1.63E-03** |

Table 6: $L^2$ error of solving PDEs on irregularly-shaped domains.

| $L^2$ Error | SKS | DAKS | PINN |
|---|---|---|---|
| Nonlinear Elliptic | 8.40E-04 | **4.86E-05** | 4.20E-02 |
| Allen-Cahn ($a = 15$) | **8.30E-02** | 6.06E-01 | 1.00E+00 |

In summary, we believe that in general the grid shape should be viewed as an influence factor in running our method, which needs to be carefully selected. The appropriate choice may also connect to the intrinsic property of the PDE itself.

### C.3 Irregularly-Shaped Domains

We tested on solving the nonlinear elliptic PDE (15) and the Allen-Cahn equation (17) with $a = 15$. For the nonlinear elliptic PDE, the domain is an inscribed circle within $[0, 1] \times [0, 1]$. For the Allen-Cahn equation, the domain is a triangle with vertices at at $(0, 0)$, $(1, 0)$ and $(0.5, 1)$. The solution is prescribed as in our paper, with boundary conditions derived from the solution. For both PDEs, our method (SKS) used a virtual grid on $[0, 1] \times [0, 1]$ that covers the domain. For DAKS and PINN, we sampled the same number of collocation points from the domain. For a fair comparison, all the methods used the same set of 192 uniformly sampled collocation points on the boundary. The error of each method is given in Table 6. The point wise error is shown in Fig. 5. As we can see, on irregularly-shaped domains, our method SKS still obtains a reasonably good accuracy for both cases (note that the Allen-Cahn case is much more challenging).

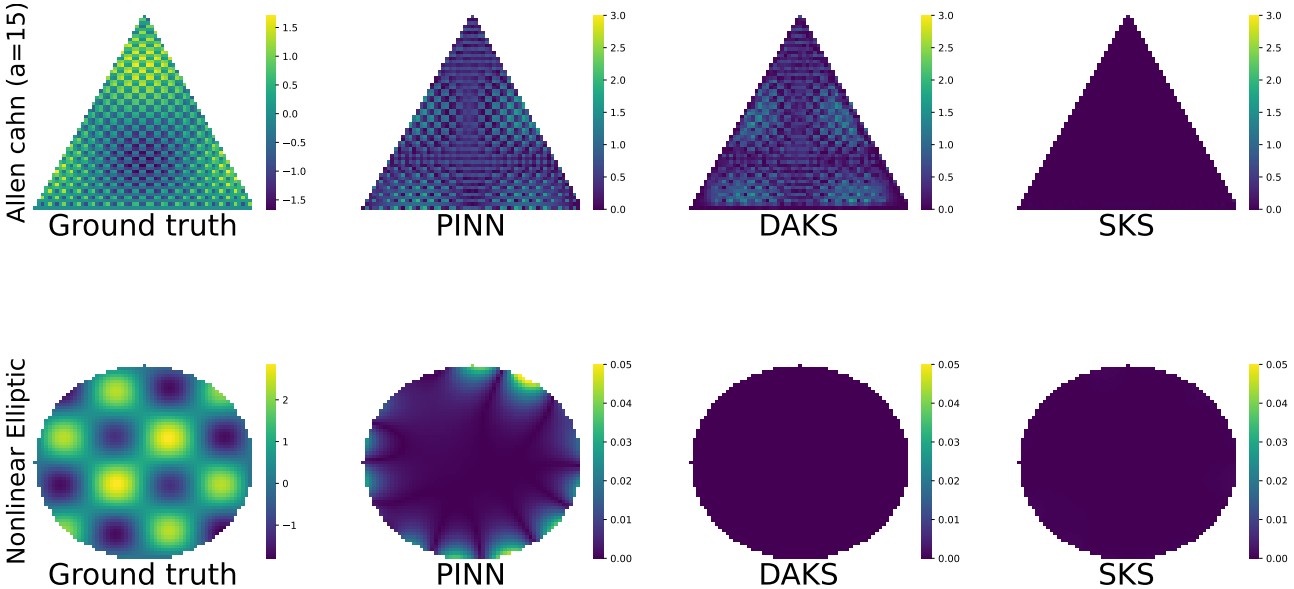

Figure 5: Point-wise solution error for Nonlinear Elliptic and 2D Allen-Cahn equation (17) with $a = 15.0$ on irregularly-shaped domains.

Table 7: $L^2$ error of SKS using different length scales.

(a) Nonlinear Elliptic PDE (15).

| Length-scale | 0.05 | 0.1 | 0.2 | 0.3 |
|---|---|---|---|---|
| $L^2$ error | 1.19E-02 | **6.80E-06** | 4.62E-05 | 9.14E-04 |

(b) Allen-Cahn Equation (17) where $a = 15$.

| Length-scale | 0.08 | 0.06 | 0.04 | 0.02 |
|---|---|---|---|---|
| $L^2$ error | 4.98E-01 | 3.83E-01 | **5.15E-03** | 2.19E-01 |

Table 8: Sensitivity of PINN to the network width and depth.

(a) $L^2$ error for depth=10 and layer width from 10 to 50.

| Layer width | 10 | 20 | 30 | 40 | 50 |
|---|---|---|---|---|---|
| Nonlinear Elliptic | 1.75E-01 | 7.68E-02 | 7.01E-02 | 3.00E-02 | 4.06E-02 |
| Allan-cahn ($a = 15$) | 6.31E+00 | 5.61E+00 | 7.08E+00 | 7.65E+00 | 1.03E+01 |

(b) $L^2$ error for depth=10 and layer width from 60 to 100.

| Layer width | 60 | 70 | 80 | 90 | 100 |
|---|---|---|---|---|---|
| Nonlinear Elliptic | 1.66E-02 | 3.58E-02 | 2.74E-02 | 2.32E-02 | 5.01E-02 |
| Allan-cahn ($a = 15$) | 1.30E+01 | 1.26E+01 | 1.20E+01 | 1.34E+01 | 1.09E+01 |

(c) $L^2$ error for layer width=10 and depth from 3 to 10.

| Depth | 3 | 5 | 8 | 10 |
|---|---|---|---|---|
| Nonlinear Elliptic | 3.78E-02 | 7.33E-02 | 7.01E-02 | 6.57E-02 |
| Allan-cahn ($a = 15$) | 1.13E+01 | 1.35E+01 | 7.08E+00 | 1.10E+01 |

(d) $L^2$ error for layer width=10 and depth from 15 to 30.

| Depth | 15 | 20 | 25 | 30 |
|---|---|---|---|---|
| Nonlinear Elliptic | 9.16E-02 | 1.13E-01 | 1.50E-01 | 7.49E-02 |
| Allan-cahn ($a = 15$) | 1.14E+00 | 9.30E+00 | 1.14E+00 | 9.35E+00 |

## C.4 Sensitivity to Hyper-Parameters

To examine the sensitivity to the choice of kernel parameters, we run SKS to solve nonlinear Elliptic PDE (15) and Allen-Cahn equation (17) with $a = 15$, with a varying set of length-scale parameters. For the nonlinear elliptic PDE, we employed the grid of size $35 \times 35$ while for the Allen-Cahan equation, we used the grid of size $49 \times 49$. The results are given in Table 7. As we can see, different length-scale parameters results in changes of orders of magnitude in the solution error. For example, switching the length-scale from 0.05 to 0.1, the $L^2$ error for solving the nonlinear elliptic PDE decreases from 1.129E-02 to 6.80E-06. Hence, our method is sensitive to the choice of the kernel parameters.

As a comparison, we also examined the sensitivity of PINN to the choice of the architectures. To this end, we fixed the depth at 8 and varied the layer width from 10 to 100, and also fixed the width at 30 while varying the depth from 3 to 30. We then tested PINN on solving the same PDEs. The solution error is reported in Table 8. One can see that when layer width is greater than 50 or the depth is beyond 8, there is no significant improvement in performance. The accuracy remains within the same magnitude with only minor variations. However, larger networks lead to substantially increased computational costs for PDE solving.

## C.5 Optimizing Basis Coefficients

To investigate the difference with RBF methods, we tested on optimizing the basis coefficients, namely $\boldsymbol{\alpha} = \mathbf{K}^{-1}\boldsymbol{\eta}$, instead of the solution values $\boldsymbol{\eta}$, for our model. We employed the same optimization approach. We tested on solving Burger's equation with $\nu = 0.02$. The results are reported in Table 10. We can see that the relative $L^2$ error increases significantly as compared to learning $\boldsymbol{\eta}$, worsening by an order of magnitude when using 1200 or 2400 collocation points, and by over 50% when using 4800 collocation points. This suggests that optimizing the coefficients $\boldsymbol{\alpha}$ is more challenging, as each

Table 9: Runtime in seconds with respect to the number of collocation points. SKS-Naive means running SKS using naive full kernel matrix computation, without leveraging the Kronecker product structure (see Section 4). Note that N/A means the method is not able to run with the corresponding number of collocation points.

(a) The Burgers' equation (14) with $\nu = 0.001$.

| Method | 2400 | 4800 | 43200 |
|---|---|---|---|
| SKS (per-iter) | **4.6E-4** | **9.8E-4** | **6.8E-3** |
| SKS-Naive (per-iter) | 1.4E-02 | 5.4E-02 | N/A |
| DAKS (per-iter) | 7.43 | 38.5 | N/A |
| PINN (per-iter) | 2.7E-1 | 5.2E-1 | 4.1E-1 |
| SKS (total) | **22.15** | **94.25** | **2007.1** |
| DAKS (total) | 89.14 | 462.18 | N/A |
| PINN (total) | 706.24 | 721.94 | 6454.7 |

(b) Allen-Cahn equation (17) with $a = 15$.

| Method | 2400 | 4800 | 6400 | 8100 | 22500 |
|---|---|---|---|---|---|
| SKS (per-iter) | **3.6E-4** | **9.1E-4** | **1.2E-3** | **1.8E-3** | **5.9E-3** |
| SKS-Naive (per-iter) | 1.1E-2 | 4.3E-2 | 7.2E-2 | 1.1E-1 | N/A |
| DAKS (per-iter) | 2.1 | 10.5 | N/A | N/A | N/A |
| PINN (per-iter) | 5.6E-2 | 1E-1 | 1.3E-1 | 1.5E-1 | 4.3E-1 |
| SKS (total) | 27.1 | 99.56 | 116.8 | 132.57 | 474.34 |
| DAKS (total) | **16.44** | **84.18** | N/A | N/A | N/A |
| PINN (total) | 2821 | 5112 | 6287 | 7614 | 21375 |

Table 10: $L^2$ error of learning $\boldsymbol{\eta}$ *vs.* $\boldsymbol{\alpha}$ on solving Burgers' equation with $\nu = 0.02$.

| Method | 1200 | 2400 | 4800 |
|---|---|---|---|
| Optimizing $\boldsymbol{\eta}$ | 5.40E-03 | 7.83E-04 | 3.21E-04 |
| Optimizing coefficients $\boldsymbol{\alpha} = \mathbf{K}^{-1}\boldsymbol{\eta}$ | 2.80E-02 | 1.56E-03 | 7.14E-04 |

coefficient exerts a strong global influence on the entire solution approximation, including both boundary conditions and internal regions. We empirically observed that conflicts often arise between fitting the boundary conditions and minimizing the PDE residuals, resulting in less effective optimization outcomes.

### C.6   Effectiveness of Regular Grid Points for DAKS

The original DAKS (Chen et al., 2021) utilized randomly sampled collocation points. In this study, we evaluated the performance of DAKS using regular grid points as collocation points to solve Burgers' equation with $\nu = 0.02$. The evaluation was conducted on exactly the same grids used in our method (SKS). As shown in Table 11, regular grid points proved less effective than random collocation points for DAKS.

## D   Limitation and Discussion

Currently, the most effective training for SKS is fulfilled by stochastic optimization, namely ADAM. We need to run a large number of ADAM epochs to achieve a promising solution accuracy. It means that the PDE solving procedure is slow. The second-order optimization methods, such as L-BFGS, neither improve the solution accuracy nor accelerate the convergence. We have also tried the relaxed Gauss-Newton approach as used in DAKS. However, this method can only achieve good performance on the nonlinear elliptic PDE, and easily diverges on the other cases. This might stem from that we take derivatives (or other linear operators) over the kernel interpolation form, which makes the convergence of the fixed point iterations used in DAKS much more difficult. We plan to develop novel Gauss-Newton relaxations to ensure convergence and stableness (at least in practice) so that we can further accelerate the PDE solving.

Table 11: $L^2$ error of different sampling methods of DAKS. Results are based on solving Burgers with $\nu = 0.02$.

| Sampling method | 600 | 1200 | 2400 | 4800 |
|---|---|---|---|---|
| Grid | 4.09E-01 | 3.85E-01 | 4.27E-02 | 5.67E-02 |
| Random | 1.75E-02 | 7.90E-03 | 8.65E-04 | 9.76E-05 |

