# OpenReview forum: "Toward Efficient Kernel-Based Solvers for Nonlinear PDEs"
_ICML.cc/2025/Conference — ICML 2025 poster_

### Official Review · Reviewer_DKwk · 2025-02-20

**Overall Recommendation:** 3

**Summary:**

This paper presents a fair contribution to kernel-based PDE solvers for nonlinear PDEs, improving upon prior methods by eliminating the need for differential operator-embedded kernels. The proposed algorithm enhances computational efficiency by leveraging Kronecker product properties and avoiding complex Gram matrices. The paper also provides convergence proofs and error rate analysis under regularity assumptions. The proposed method is evaluated on Burgers’, nonlinear elliptic, Eikonal, and Allen-Cahn equations, showing its comparable accuracy and improved scalability compared to the baselines.

## update after rebuttal
Thank you for addressing the concerns. I am keeping my original score.

**Claims And Evidence:**

Claims are largely supported by theoretical derivations and experiments.

**Essential References Not Discussed:**

N/A

**Experimental Designs Or Analyses:**

- Strengths: The experimental setup is in general well-designed. It covers multiple PDE formulations, varying levels of difficulty (small vs. large collocation points), and different kernel settings.
- Weaknesses: The hyperparameter selection process (e.g., kernel length scales) is not discussed in detail. I'm also curious how the runtime of the Kronecker based method compares with the naive full matrix method.

**Methods And Evaluation Criteria:**

- Strengths: The methods and evaluation criteria are in general well-aligned with PDE-solving tasks. The benchmarks involve multiple nonlinear PDE formulations, and the baselines include DAKS, PINNs, and legacy finite difference methods too. The paper also offers scalability analysis, which is appreciated.
- Weakness: The method assumes a structured grid, which limits its generalization to unstructured meshes and other complex discretization routines.

**Other Comments Or Suggestions:**

N/A

**Other Strengths And Weaknesses:**

N/A

**Questions For Authors:**

See the "Weaknesses" items above.

**Relation To Broader Scientific Literature:**

This paper is built upon kernel-based PDE solvers and relates to Gaussian Process models for PDE solving. The Kronecker product approach also aligns with structured kernel methods. There is sufficient technical contribution.

**Theoretical Claims:**

This paper provides convergence analysis, proving that the method maintains error bounds similar to prior kernel PDE solvers despite using a smaller model space. Other claims are also briefly reviewed and look good, but not checked in detail.

---

> ### Author Rebuttal · Authors · 2025-03-30
>
> Thanks for your valuable and constructive comments.
>
> >The method assumes a structured grid, which limits its generalization to unstructured meshes and other complex discretization routines.
>
> R1: We appreciate your insightful comments. Indeed, we agree that our method relies on a structured grid and does not directly apply to more complex discretizations. A simple yet effective workaround — which we have validated (see Section 6 and Appendix Section C.3) — is to employ a virtual grid that covers the irregular domain.
>
> In future work, we plan to explore two directions to better generalize our method. First, we will investigate domain decomposition and hierarchical grid construction to adaptively adjust the grid resolution across local regions while preserving computational efficiency. Second, we aim to learn a mapping from unstructured mesh points to a latent grid, where our efficient kernel solvers can be applied. This approach is inspired by recent work such as GeoFNO[1] in the neural operator literature.
>
> We will include this discussion in the paper.
>
> [1] Li, Z., Huang, D. Z., Liu, B., & Anandkumar, A. (2023). Fourier neural operator with learned deformations for pdes on general geometries. Journal of Machine Learning Research, 24(388), 1-26.
>
> >The hyperparameter selection process (e.g., kernel length scales) is not discussed in detail.
>
> R2: Thank you for your great suggestion. Given the wide range of hyperparameters --- including $\alpha$ and $\beta$ (see our response R3 to Reviewer fQAE), as well as kernel length-scales and nugget values (see Lines 284–297 right column) --- performing a full grid search would be prohibitively expensive. Therefore, we adopt a hybrid strategy. We begin with a random search to identify a promising set of hyperparameters. Then, we perform a grid search over $\alpha$ and $\beta$, keeping the other parameters fixed. Once $\alpha$ and $\beta$ are selected, we fix them and conduct a grid search over the remaining hyperparameters, including the nugget and length-scales. We will include a detailed description of this procedure in the paper.
>
> >I'm also curious how the runtime of the Kronecker based method compares with the naive full matrix method.
>
> R2: Great question. Below, we provide the runtime of our model using the naive full matrix computation (i.e., without exploiting the Kronecker product structure). As shown, the per-iteration runtime with naive matrix operations is consistently around **100× slower** compared to our method, which leverages Kronecker product properties. Furthermore, when the number of collocation points increases to 22,500 for the Burgers’ equation and 43,200 for the Allen–Cahn equation, the naive approach exceeds available memory, resulting in out-of-memory (OOM) errors — rendering it infeasible to run. We will include these results in our paper.
>
> | 	        Allen-Cahn ($a=15$)              | 2400     	 | 4800     	 | 6400     	 | 8100    | 22500 |
> |-------------------------------------------|------------|------------|-----------|---------|-------|
> | **Naive computation** (per-iter)            	 | 1.1E-02 	  | 4.3E-02 	  | 7.2E-02   | 1.1E-01 | OOM   |
> | SKS (per-iter)            	               | 3.6E-4     | 9.1E-4     | 1.2E-3    | 1.8E-3  | 5.9E-3  |
> | DAKS (per-iter)            	              | 2.1        | 10.5       | OOM          | OOM     | OOM     |
> | PINN (per-iter)            	              | 5.6E-2     | 1E-1       | 1.3E-1    | 1.5E-1  | 4.3E-1  |
>
> | 	  Burgers' $\nu=0.001$                  | 2400     	 | 4800     	 | 43200     	 |
> |------------------------------------------|------------|------------|-------------|
> | **Naive computation** (per-iter)           	 | 1.4E-02 	  | 5.4E-02 	  | OOM         |
> | SKS (per-iter)           	               | 4.6E-04 	  | 9.8E-04 	  | 6.8E-03     |
> | DAKS (per-iter)           	              | 7.43 	     | 38.5 	     | OOM         |
> | PINN (per-iter)           	              | 2.7E-01 	  | 5.2E-01 	  | 4.1E-01     |

---

### Official Review · Reviewer_fonM · 2025-03-10

**Overall Recommendation:** 3

**Summary:**

This paper introduces a novel kernel learning framework for efficiently solving nonlinear partial differential equations (PDEs). Unlike existing methods that embed differential operators within kernels, this approach eliminates these operators from the kernel, using standard kernel interpolation to model the solution. By differentiating the interpolant, the method avoids the need for complex Gram matrix construction, which simplifies implementation and enables efficient computation. The framework leverages Kronecker product structures for scalable computation, allowing it to handle large numbers of collocation points. The authors provide a rigorous convergence analysis and demonstrate the method's efficacy on several benchmark PDEs, showing competitive performance and scalability, particularly in challenging scenarios requiring dense grids.

**Claims And Evidence:**

The main claim is that the newly proposed method has better computation efficiency and scalability, which are well supported by the experiments.

**Essential References Not Discussed:**

No

**Experimental Designs Or Analyses:**

The description and analyses of experiments are valid.

**Methods And Evaluation Criteria:**

The kernel method and PDE solution accuracy criteria are suitable.

**Other Comments Or Suggestions:**

Readability can be improved by adding: 1）intuitive and informal description of math derivations in Sections 2 and 3,  2) bullets in Section 5.

**Other Strengths And Weaknesses:**

The main strength of the proposed method is the acceleration of kernel-based PDE solvers.

**Questions For Authors:**

1. Can you provide more details on the virtual grid on an irregular domain, which is important for the application of your method?
2. Can you provide the running time comparison with the finite difference method?
3. In future work, can the proposed method be extended to the neural operator setting for further acceleration?

**Relation To Broader Scientific Literature:**

Kernel Method, Numerical method of PDEs.

**Theoretical Claims:**

I roughly checked the correctness of theoretical claims.

---

> ### Author Rebuttal · Authors · 2025-03-30
>
> We thank the reviewer for the valuable and constructive comments.
>
> >C1: Can you provide more details on the virtual grid on an irregular domain, which is important for the application of your method?
>
> R1: Great question. Our virtual grid is chose as the smallest rectangular grid that fully covers the irregular domain. Specifically,  for the  nonlinear elliptic PDE (see Appendix C.3 and Figure 4), the domain is a circle inscribed within $[0, 1]\times [0, 1]$, and we hence set the virtual grid on $[0, 1]\times [0, 1]$ with a grid size of **$50 \times 50$**. For the Allen-Cahn equation (see Appendix Figure 4), the domain is a triangle with vertices at (0, 0), (1,0) and (0.5, 1). To fully cover this domain, we again place the virtual grid on $[0, 1] \times [0, 1]$, using a grid size of $50 \times 50$. We will include these details in our paper.
>
> > C2: Can you provide the running time comparison with the finite difference method?
>
> R2: Great suggestion. Below, we provide the time comparison results in seconds (for the Allen-chan equation with $a=15$).
>
> | 	                           | 2400     	 | 4800     	 | 6400     	 | 8100    | 22500   |
> |-----------------------------|------------|------------|------------|---------|---------|
> | **FD (per-iter)**             	 | 1.6E-02 	  | 1.4E-02 	  | 1.4E-02    | 1.4E-02 | 1.5E-02 |
> | SKS (per-iter)              | 3.6E-4     | 9.1E-4     | 1.2E-3     | 1.8E-3  | 5.9E-3  |
> | DAKS (per-iter)             | 2.1        | 10.5       | N/A        | N/A     | N/A     |
> | PINN (per-iter)             | 5.6E-2     | 1E-1       | 1.3E-1     | 1.5E-1  | 4.3E-1  |
> |    |      |        |      |      |      |
> | **FD (total)**	                 | 0.13 	     | 0.14 	     | 0.16 	     | 0.18    | 1.15    |
> | SKS (total)	                | 27.1 	     | 99.56 	    | 116.8 	    | 132.57  | 474.34  |
> | DAKS (total)	               | 16.44 	    | 84.18 	    | N/A 	      | N/A     | N/A     |
> | PINN (total)	               | 2821 	     | 5112 	     | 6287 	     | 7614    | 21375   |
>
> As we can see, the finite difference (FD) is computationally much more costly per iteration compared to our method (SKS). This might be due to the expensive cost of computing the inverse Jacobian during the root finding procedure. However, FD typically converges within just a few dozen iterations — substantially faster than the stochastic optimization used in both SKS and PINN — leading to a much lower total runtime. We will include this comparison in the paper.
>
> > C3: In future work, can the proposed method be extended to the neural operator setting for further acceleration?
>
> R3: We appreciate the reviewer for bringing up this excellent idea — we completely agree. In fact, we have already begun exploring this direction in our ongoing work. One line of effort involves replacing the trunk network in the Deep Operator Network (DeepONet) with Gaussian process bases. This enables us to adopt a similar computational strategy to accelerate both training and prediction, while also providing a natural framework for uncertainty quantification. We are also extending our approach to deep kernel-based operator learning, where our method can further accelerate function transformations in the latent channel space. We look forward to continuing our exploration in this promising direction.

---

### Official Review · Reviewer_fQAE · 2025-03-19

**Overall Recommendation:** 4

**Summary:**

The paper proposes a new twist to a kernel-based solver for PDEs. It builds on previous work by relaxing the constraints associated to the PDE to facilitate optimization. By placing the collocation points on a grid and using a decomposable kernel, the inversion of a large matrix is broken down into many inversions of small matrices, which improves the performance. A convergence analysis is provided, showing that this solver enjoys similar properties as previous work i.e. the Sobolev norm of the residuals goes to zeros as the grid uniformly covers the input space. Numerical experiments highlight the benefits of the approach and compare to competitors (PINNs and the former kernel-based solver).

## update after rebuttal

The authors clarified my question about how the constraint was handled using lagrangian duality. This and the rebuttal to other reviews convinced me to increase my score.

**Claims And Evidence:**

Yes, the claims are valid and supported by evidence, both theoretical and empirical. The performances are not uniformly superior to competitors but I do not think it is an issue. The wording in the conclusion ("encouraging performances") might be closer to reality that within the abstract ("demonstrate the advantage").

**Essential References Not Discussed:**

I would have liked to see discussed [Learning Partiel Differential Equations in Reproducing Kernel Hilbert Spaces, George Stepaniants, JMLR 2023] and references therein. Overall I think that the overview in the introduction lacks references about related approaches beyond the scope of [Chen et al. 2021].

**Experimental Designs Or Analyses:**

I have no issue with the experimental design.

**Methods And Evaluation Criteria:**

Yes, the methods make sense. They seem to be standard in this area of research and have been investigated in other papers.

**Other Comments Or Suggestions:**

I think that there is a problem with the formulation of (7). As it is, $\epsilon$ has no impact on the problem thus I hardly see how it can be equivalent to (6). The regularization term should grow as $(P(u)(x) - f(x))^2$ is away from $[0, \epsilon]$. A typical way to do this is to square this quantity again.

**Other Strengths And Weaknesses:**

+: The paper is relevant to the ICML community.

-: The introduction lacks references. Only the direct relevant work is cited (2 papers) which makes it hard to situate the paper at first.

-: There are no standard deviations on the performance metrics

**Questions For Authors:**

The proposed formulation for handling the constraints relies on two additional regularization parameters. While you demonstrate the equivalence between formulations (6) and (7), this is only true for the perfect regularization strength, which we do not know. Could you include a discussion about the tuning of these parameters ?

**Relation To Broader Scientific Literature:**

The paper proposes a novel way to encode the constraints for solving PDEs, improving previous work. The convergence analysis is similar to what was done for previous work as well. Overall, the paper is very close to their main source of inspiration.

**Theoretical Claims:**

To the best of my knowedge, the theoretical claims are valid.

---

> ### Author Rebuttal · Authors · 2025-03-30
>
> Thanks for your valuable and insightful comments.
>
> >The introduction lacks references. Only the direct relevant work is cited (2 papers)
>
> R1: Thank you for the helpful suggestion. We will include additional references in the introduction to provide broader context and better situate our work within the existing literature from the outset.
>
> >no standard deviations
>
> R2: Great comment. In our experiments, we observed that despite using stochastic optimization (ADAM), our method (SKS) consistently converge to the same solution. So is PINN. We ran multiple trials on each PDE and found that the standard deviation of the error  is negligible. For example, below we show  the standard deviation of the $L^2$ error for our method when solving Burgers' equation ($\nu = 0.01$) and the nonlinear elliptic equation. Given the extremely low variance, we chose to omit standard deviation values from the main results (Note that the finite difference method is deterministic and does not exhibit variance). We will clarify this point in our paper.
>
> Burgers($\nu = 0.01$):
>
> | 600                  	| 1200                 	| 2400                 	| 4800            	|
> |---------------------|----------------------|----------------------|-----------------|
> | 1.44E-02 $\pm$ 1.52E-07 	| 5.40E-03 $\pm$ 1.39E-07 	| 1.12E-03 $\pm$ 7.76E-08 	| 3.21E-04 $\pm$ 0.0 	|
>
> Nonlinear elliptic:
>
> | 300             	| 600             	| 1200            	| 2400            	|
> |-----------------|-----------------|-----------------|-----------------|
> | 1.26E-02 $\pm$ 0.0 	| 6.93E-05 $\pm$ 0.0 	| 6.80E-06 $\pm$ 0.0 	| 1.83E-06 $\pm$ 0.0 	|
>
> >I think that there is a problem with the formulation of (7). As it is, $\epsilon$  has no impact on the problem thus I hardly see how it can be equivalent to (6)... A typical way to do this is to square this quantity again.
>
> R2: Thank you for your insightful question. Here is our clarification.
>
> First, **$\epsilon$ plays an important theoretical role** (though in practice, it can often be set to zero). Our convergence proof and rate estimate (see Lemma 4.2 and Appendix Section A) are established by systematically varying $\epsilon$. Specifically, we set $\epsilon = C_0 h^{2\tau}$ and vary $\epsilon$ by adjusting the fill-in distance $h$. Each choice of $\epsilon$ defines a distinct instance of problem (6), with its own solution. We prove that as $\epsilon \to 0$ (i.e., $h \to 0$), the solution of (6) converges to the ground-truth solution of the PDE (see Appendix Section A for details).
>
> Second, to show that solving (7) (with appropriately chosen $\alpha$ and $\beta$) is equivalent to solving (6), we use the **Lagrangian formulation** of (6). It is a standard result that constrained optimization problems can be reformulated as **mini-max problems over the Lagrangian**. The Lagrangian is a *linear* combination of the objective and the constraints — it does **not** involve squaring the constraints. From this perspective, it is straightforward to show that when $\alpha$ and $\beta$ are selected as the optimal dual variables in the mini-max problem, solving (7) yields the optimal $u$, and thus is equivalent to solving (6). This equivalence holds for **any** $\epsilon$, not just in the limit as $\epsilon \to 0$. The full proof is provided in Appendix Section B.
>
> Finally, we agree that an alternative approach --- minimizing the objective plus the squared constraints (i.e., a penalty method) --- is also a good and viable idea. However, this method is primarily justified for equality constraints (i.e., $\epsilon = 0$) (While it is possible to design penalty terms for inequality constraints, they typically introduce non-differentiable terms.) Moreover, equivalence to the original constrained problem is only guaranteed as as **the penalty weights tend to infinity**. Therefore, we believe our formulation in (7) offers stronger theoretical guarantees and greater practical flexibility when approximating solutions of (6). We will include a more detailed discussion of this alternative approach in our paper.
>
> >Could you include a discussion about the tuning of two regularization parameters (7)?
>
> R3: Thank you for the great suggestion — we completely agree. In our experiments, we selected $\alpha$ and $\beta$ in Equation (7) from a wide range:
> $[10^{-2}, 10^{-1}, 1, 10, 10^2, 10^3, 10^4,10^5, 10^6, 10^7, 10^8, 10^{10}, 10^{12}, 10^{14}, 10^{15}, 10^{20}]$,
> jointly with other hyperparameters, including the kernel length scales and nugget terms (see their ranges in Lines 284-297 right). To efficiently tune hyperparameters, we first performed a random search to identify a promising group of hyperparameters. We then fixed all other hyperparameters and conducted a grid search over $\alpha$ and $\beta$. Finally, we fixed $\alpha$ and $\beta$ and performed a grid search over the remaining hyperparameters. We will include this discussion in the paper.

---

> > ### Comment · Reviewer_fQAE · 2025-04-07
> >
> > I would like to thank the authors for their response.
> >
> > Concerning the formulation of (7): thanks for the clear explanation, my comment was mislead.
> >
> > After reading the other reviews and their rebuttals, I do not think that there are major issues with the paper. It is sound work, and as a reader I would be happy to see it among the ICML papers this year. I moved my score up a bit to reflect that.

---

> > > ### Author Response · Authors · 2025-04-08
> > >
> > > Thank you for your response. We appreciate your support and positive feedback!

---

### Official Review · Reviewer_P51e · 2025-04-03

**Overall Recommendation:** 3

**Summary:**

The authors propose an assymetric RBF collocation method for solving general PDEs from a Gaussian process/RKHS point of view. They parametrize the solution $u(x;\eta)$ as a (Gaussian RBF) kernel interpolant given function values on collocation points, then differentiate this representation to obtain derivatives values at collocation points, which they then optimize to approximately satisfy the PDE on collocation points using a least squares formulation.

They then take advantage of the fact that under the assumption that the kernel $k$ factorizes across different dimensions of the input space, then both $\eta \to u(x;\eta)$ aand $\eta \to Lu(x;\eta)$ posses efficient efficient evaluation formulas via Kronecker-factored matrix vector products.

With the parametrization and fast evaluation in hand, their algorithm consists of minimizing a regularized squared residual of PDE mismatch on interior collocation points and boundary mismatch using stochastic optimization (ADAM).

**Claims And Evidence:**

The claims made in the paper seem well substantiated by evidence. I am only unsure as to why SKS would outperform DAKS--the theory for DAKS seems stronger, but a good implementation is difficult, and especially with the RBF kernel, matrices can get very ill-conditioned, making it hard to disambiguate discretization error from numerical roundoff error.

**Essential References Not Discussed:**

None that I know of.

**Experimental Designs Or Analyses:**

The experiments seem valid

**Methods And Evaluation Criteria:**

The problems and evaluation criteria seem good. I am a little bit skeptical of ADAM being the best solver for the problem, compared to a quasi-Newton or Newton-Krylov method, but their experiments corroborate this choice.

**Other Comments Or Suggestions:**

The term "Gaussian-Newton" seems odd to refer to "Gauss-Newton" methods.

The implementation on irregular domains is a bit unclear. Throughout the rest of the paper, it is assumed that the basis points are the same as collocation points, but this seems hard to do with irregular domains (especially for the boundary. This should be clarified.

The authors should include an error plot as a function of the mesh norm $h$ to show the order of convergence of the algorithm (using Gaussian RBF should theoretically give spectral convergence?).

The authors should clarify assumption C3 in relation to their choice of Gaussian RBF kernel. Is it true that the RKHS associated to a Gaussian RBF kernel can be continuously embedded in $H^k$ for every $k$?

**Other Strengths And Weaknesses:**

The strengths include a nice analysis of feasibility for the least squares problem, and a simple algorithm for general PDEs.

The sensitivity to hyperparameters seems to be a weakness (lengthscale and nugget).

**Questions For Authors:**

What was the batch size and hyperparameters used in the stochastic optimization?

What did the results with attempting to apply LBFGS to improve the solutions look like? In the PINN literature, it seems that it often significantly improves the solution after running ADAM to warm up.

Did the gradient norms of the objective go to zero at the end (did you reach a minimum, or stall out due to ill conditioning)?

**Relation To Broader Scientific Literature:**

The main contributions are as follows:
1. Convergence analysis of least squares methods for assymetric collocation--a bound on the attainable squared residual norm, and convergence rates of such least squares solutions to a true solution of the PDE.
2. Taking advantage of the Kronecker factored structure of the kernel matrices when the kernels factorize to obtain fast evaluation formulas and efficient optimization
3. Interpretation of RBF collocation methods as optimization over function values that are interpolated rather than coefficients of an expansion.

**Theoretical Claims:**

The proofs look sound. I read the first part of the proof of Lemma 4.2 carefully, and it looks all correct except for a minor typo in equation (22) which seems like its should be $u_m^*$ on the left.

---

### Decision · Program_Chairs · 2025-05-01

**Decision:**

Accept (poster)

**Comment:**

The paper introduces a kernel learning framework for solving nonlinear partial differential equations efficiently. The key idea is to pose the problem as a kernel interpolant by relaxing the constraints. By placing the collocation points on a grid and using a decomposable kernel, the inversion of a large matrix is broken down into many inversions of small matrices, which improves the performance. Theoretical and experimental results are presented to demonstrate the efficacy of the proposed approach.

All the reviewers are positive about the paper, and the contribution of a fast kernel learning framework is interesting and useful. The rebuttal to the authors' comments appears satisfactory. I strongly urge the authors to incorporate the reviewer comments and the rebuttal discussion in the camera-ready version. Based on the reviewer comments, I propose to accept the paper.